# An RNA helicase coordinates with iron signal regulators to alleviate chilling stress in *Arabidopsis*

Yingying Xing ®[1], Yawen Li[2], Xinmeng Gui[3], Xianyu Zhang[1], Qian Hu[1], Qiqi Zhao[1], Yongli Qiao ®[3], Ning Xu ®[1] ✉ & Jun Liu ®[1] ✉

Chilling stress is one of the major environmental stresses that restrains plant development and growth. Our previous study showed that a potential iron sensor BTS (BRUTUS) was involved in temperature response in Arabidopsis plants. However, whether plant iron homeostasis is involved in plant response to temperature fluctuation is not known. In this study, we discover that *BTS* mutant *bts-2* is sensitive to chilling stress, and the sensitivity is attributed to the accumulation of iron. The suppressor screening of *bts-2* led to the discovery of RH24, a DEAD-box RNA helicase, that fully suppresses *bts-2* chilling sensitivity. RH24 is accumulated under low temperatures, where it unwinds the iron regulator *ILR3* (IAA-leucine resistant 3) mRNA and increases the ILR3 protein levels. Intriguingly, RH24 sequesters ILR3 in phase-separated condensates to reduce ILR3-mediated iron overload, and BTS or cold treatments further facilitated the condensate formation. Therefore, RH24 and BTS coordinately control ILR3 to reduce iron uptake under chilling stress. Our findings reveal that the RNA helicase RH24 and BTS finetunes ILR3 to maintain plant iron homeostasis in response to temperature fluctuations.

Chilling stress, as one of the major environmental stresses, restrains plant growth, development, and geographic distribution. Due to global climate change, extreme low-temperature events frequently occur, resulting in enormous economic losses in agriculture[1]. For the chilling-sensitive plants, chilling stress usually impairs their growth and development at both vegetative and reproductive stages, and is often accompanied by stunted growth, decreased germination, leaf chlorosis, and even ultimate death of the plants[2,3]. To survive under low temperatures, plants have evolved complex regulatory mechanisms to ameliorate the stress caused by low temperatures[4–6].

Chilling stress is known to reduce cell membrane fluidity, change the redox state, inhibit photosynthesis and enzyme activities, and often leads to accumulated reactive oxygen species and elevated electrolyte leakage[7,8]. Typical transcription factors, such as C-repeat

binding factor (*CBFs*) genes, were rapidly induced under chilling conditions. These induced CBFs directly activate a set of downstream cold-regulated (*COR*) genes, which consequently enhance plant chilling tolerance[1,9–11]. Despite the essential roles of CBFs in plant chilling tolerance, only about 10% of *COR* genes are regulated by CBFs. In fact, even the high order of *cbfs* mutants still have considerable tolerance to cold temperatures, suggesting that CBF-independent signaling pathways are involved in regulating cold tolerance in plants[12,13].

In addition to the transcription factors, post-transcriptional regulation of mRNA is also involved in plant chilling response[14–16]. RNA helicase is one of the key protein families that control the post-transcriptional regulation of mRNA. RNA helicases are the enzymes that ensure the formation of mature RNAs in the correct structure

[1]State Key Laboratory of Agricultural and Forestry Biosecurity, MOA Key Lab of Pest Monitoring and Green Management, College of Plant Protection, China Agricultural University, Beijing, China. [2]State Key Laboratory of Plant Genomics, Institute of Microbiology, Chinese Academy of Sciences, Beijing, China. [3]Shanghai Key Laboratory of Plant Molecular Sciences, College of Life Sciences, Shanghai Normal University, Shanghai, China. ✉e-mail: xn741852@163.com; Junliu@im.ac.cn

via their RNA-unwinding and RNA-unfolding activities[17]. DEAD (Asp-Glu-Ala-Asp)-box RNA helicase is the largest class of RNA helicase family. It contains nine conserved motifs in the helicase core domain[18,19]. The helicase core domain is required for RNA and ATP binding, and is essential for hydrolyzing mRNA[20]. In addition, DEAD-box RNA helicase contains the variable extension sequences at its N-and C-termini, and is proposed to define substrate binding specificity[18,21]. Thus, each DEAD-box RNA helicase plays a unique role in RNA metabolism, including mRNA splicing, export, translation, and RNA processing[19,22].

It has been reported that many DEAD-box RNA helicases act in plant chilling responses in CBF-independent pathways. For example, RH42/RCF1 maintains proper pre-mRNA splicing to regulate cold tolerance in plants[23,24]. RH25 is induced by cold stress, and over-expression of RH25 enhances the plant freezing tolerance[25]. RH38 plays an essential role in mRNA export and is involved in plant cold stress response[14,26]. RH7/PRH75 participates in 18S pre-rRNA processing when the plants are subjected to cold stress[15,27]. These lines of evidence demonstrate that DEAD-box RNA helicases diversely assist plants in adapting to low-temperature stress.

Iron (Fe) is the most important microelement for almost all organisms. It is a cofactor for many key enzymes in diverse cellular processes, including chlorophyll biosynthesis, respiration, DNA replication, and energy production[28,29]. Iron deficiency can lead to anemia in humans and chlorosis in plants, but overloading of iron can be harmful due to the induction of the Fenton reaction, which produces hydroxyl radicals to damage DNA and peroxidize proteins or lipids[29]. Because of the importance of iron, maintaining iron homeostasis under environmental stresses is critical for plant survival[30–32]. Comparative transcriptome analysis of rice under cold, iron, and salt stresses revealed that 468 differential expressed genes (DEGs) were found under both cold and iron stresses[33]. FRO2 (ferric reduction oxidase 2), the ferric-chelate reductase which reduces iron from $Fe^{3+}$ to $Fe^{2+}$ for iron uptake, is required for glycine betaine-induced chilling tolerance in Arabidopsis roots[34,35]. In tobacco, the accumulation of ferritin in the chloroplast protects the plant against low temperature-induced photoinhibition[36]. The increased iron accumulation in Arabidopsis roots is also required for the chilling tolerance associated with phosphate (Pi) starvation[37]. Low temperature reduces photosynthesis efficiency in plants and damages photosystem I (PSI). PSI system contains iron, and chilling stress can affect iron homeostasis by impacting PSI function[37]. These evidences imply that iron homeostasis is highly related to plant chilling tolerance. Nevertheless, how do plants regulate iron homeostasis to enhance chilling tolerance remains unclear.

The potential iron sensor protein BRUTUS (BTS) is a key regulator of iron homeostasis. It regulates iron deficiency response by promoting the degradation of subgroup basic helix-loop-helix (bHLH) transcription factors ILR3 (IAA-leucine resistant 3) and bHLH115[38]. Under iron deprivation, ILR3 is activated, which then facilitates iron uptake and storage in plants[39,40]. However, BTS targets and degrades ILR3 to prevent plants from acquiring excessive iron.

Our previous study showed that *bts-2* is temperature sensitive[41]. In this study, we resolved the question of why the BTS mutant *bts-2* is sensitive to low temperatures and found that overloading of iron in plants reduced their chilling tolerance. We discovered a DEAD-box RNA helicase, RH24, that could fully suppress the chilling sensitivity and iron stress tolerance phenotypes of *bts-2*. RH24 is accumulated under chilling stress. It unwinds *ILR3* mRNA to increase ILR3 protein levels. However, RH24 and ILR3 form phase-separated condensates under chilling stress, and BTS facilitates the condensate formation. As a result, plants reduce iron uptake. We propose that RH24 and BTS precisely modulate iron homeostasis via ILR3 to alleviate chilling stress in Arabidopsis.

## Results

### Iron accumulation increases the chilling sensitivity of plants

BTS (BRUTUS) is a key regulator of iron homeostasis, in which it negatively regulates iron deficiency responses by facilitating the degradation of a set of downstream bHLH transcription factors[38]. *BTS* null mutant (*bts-2*) accumulates a higher concentration of iron and therefore is resistant to iron deficiency[38]. Previously, we showed that the *bts-2* mutant was chilling sensitive, which displayed a chlorotic and stunted phenotype under standard growth conditions but was rescued by growing at 26 °C[41]. To investigate the chilling sensitivity of *bts-2*, the plants were treated at different temperatures. The result showed that *bts-2* mutants were sensitive to the temperatures of 22 °C, 8 °C, and 4 °C, revealed by lower chlorophyll concentration, increased ion leakage, and higher levels of hydrogen peroxide ($H_2O_2$) than that of Col-0 (wild type, WT) (Supplementary Fig. 1a–d). Meanwhile, *bts-2* mutants pre-cultured at 26 °C still exhibited chilling sensitivity when they were transferred to low temperatures (Supplementary Fig. 1e). Because BTS is a key regulator of Fe deficiency responses, we hypothesized that iron homeostasis is likely associated with its low-temperature sensitivity.

To examine the above assumption, we first examined the transcriptome in WT plants when they were subjected to chilling stress (4 °C), with particular attention on iron-regulated genes reported by Kim et al.[42]. The results showed that the differentially expressed genes (DEGs) exhibited remarkably reduced expression for the genes involved in regulating low-iron-inducible iron homeostasis, encoding iron deficiency-induced transcription factors and low-iron-inducible ferric chelate reductase at 4 °C in Col-0 plants. In contrast, iron deficiency-repressed genes and iron storage-related genes (*FERs*) were strongly induced at 4 °C compared with that at 22 °C (Fig. 1a), suggesting that iron homeostasis was disrupted by chilling stress. We sequenced the transcriptome of *bts-2* under chilling stress to investigate if CBFs pathways were activated and examined the differences for Fe-responsive genes in *bts-2* mutant. The results showed that the expression profile of the key cold-responsive genes that are regulated by CBF pathway[13] in *bts-2* did not have significant differences between Col-0 and *bts-2* by treatments at 4 °C for 3 h and 24 h (Supplementary Fig. 1f, g and Supplementary Data 1), suggesting that BTS-mediated chilling tolerance is largely independent of CBFs. Meanwhile, the expression of iron man peptides genes (*IMAs*) and bHLH transcription factors was significantly down-regulated in *bts-2* compared with that in Col-0 at 22 °C. However, the key iron storage-related gene *FER1* in *bts-2* were strongly induced at 4 °C, suggesting that chilling stress led to iron accumulation in *bts-2* (Fig. 1a).

To verify the transcriptome data, we then measured the iron concentration of plants grown under different temperatures. Because *bts-2* cannot survive at 4 °C (Fig. 1b), and it seemed that 8°C was the lowest temperature that *bts-2* could survive (Supplementary Fig. 1a), we then treated the plants at 8 °C, 22 °C, and 26 °C. The iron concentration of Col-0 grown under chilling treatment (8 °C) was slightly but significantly increased than the plants grown under high temperature (26 °C) (Fig. 1c). However, *bts-2* mutants accumulated much higher iron levels at standard growth conditions (22 °C) than that at 26 °C, and chilling treatment further increased iron accumulation (Fig. 1c), indicating that chilling treatment could elevate iron concentration in plants.

### Oversupply of iron undermines the plant chilling tolerance

As temperature treatments altered plant iron metabolism and endogenous iron levels (Fig. 1a, c), we then examined the chilling sensitivity of *bts-2* mutants grown on Fe-depleted, Fe-replete, and Fe-excess media. The result showed that *bts-2* mutants displayed increased chilling sensitivity following the increment of iron in the media (Fig. 1d). By contrast, the chlorophyll concentration of *bts-2* grown on Fe-replete and Fe-excess media at 8 °C were about 76.9% and 87.6%

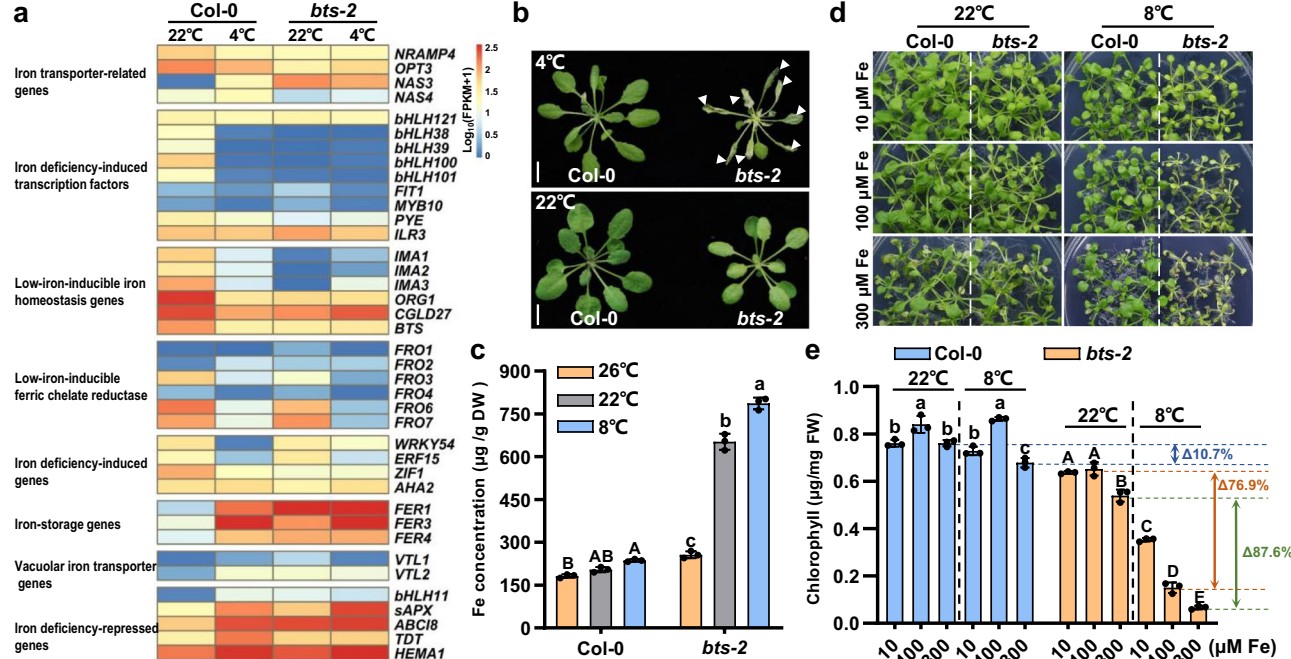

**Fig. 1 | The chilling sensitivity of *bts-2* is associated with the accumulation of iron. a** The differentially expressed genes (DEGs) of iron metabolism pathways in plants under 4 °C. Col-0 and *bts-2* plants were grown at 22 °C for 4 weeks and then were subjected to 4 °C for 24 h. The leaves of four-week-old plants were sampled for transcriptome assays. Two biological replicates were performed. Iron metabolism genes referred to Kim et al.[42]. **b** The phenotype of *bts-2* plants under chilling stress. Col-0 and *bts-2* plants were grown at 26 °C for 3 weeks, then the plants were further grown at 22 °C or 4 °C for 7 days. Arrows indicate the collapsed leaves. Bar = 1 cm. **c** Iron concentration in Col-0 and *bts-2* plants. The plants were grown in soil at 22 °C for 2 weeks and then transferred to 26, 22, and 8 °C for another 2 weeks. Fe concentration is on the dry weight basis, and the leaves of four-week-old plants were sampled for iron elemental analysis. About twenty plants were used for each replicate. Different letters indicate the significant differences based on a two-factor

ANOVA with Tukey's HSD test ($P \le 0.05$, Data are means ± SD, $n = 3$ biological replicates). **d** Oversupply of iron increased the chilling sensitivity of *bts-2*. Col-0 and *bts-2* plants were grown on ½ MS at 22 °C for 2 weeks and then transferred to Fe deprivation ½ MS medium (-Fe) supplemented with 10, 100, and 300 μM Fe(II)-EDTA for 1 week at 22 °C and 2 weeks at 8 °C, respectively. 10, 100, and 300 μM Fe correspond to Fe-depleted, Fe-replete, and Fe-excess conditions, respectively. **e** Chlorophyll concentration of Col-0 and *bts-2* seedlings in (**d**). Six plants for per replicate were collected and used for chlorophyll concentration measurement. Different letters indicate the significant differences based on a two-factor ANOVA with Tukey's HSD test ($P \le 0.05$, Data are means ± SD, $n = 3$ biological replicates). Delta values refer to the percentage decline of chlorophyll concentration at 8 °C compared to 22 °C under Fe-replete (100 μM Fe) or Fe-excess (300 μM Fe) conditions. Source data are provided as a Source Data file.

decrease than that grown at 22 °C; however, Col-0 showed chlorophyll concentration decrease only under Fe-excess media at 8 °C (decreased by 10.7%) (Fig. 1d, e). These results demonstrate that the high concentration of iron in the plants indeed impaired their chilling tolerance, especially *bts-2* plants. To reinforce this notion, we supplied exogenous iron to the Col-0 plants when they were subjected to chilling stress. Intriguingly, many leaves collapsed by 200 μM iron treatment at 4 °C (Supplementary Fig. 1h). These data further support the notion that iron accumulation impairs plant chilling tolerance.

## The *bts-2* suppressor *bts-r* rescues its chilling sensitivity

To explore how plants regulate iron hemostasis to adapt to temperature fluctuations, we attempted to identify the gene(s) that could suppress the *bts-2* chilling-sensitive phenotype. Over 4000 *bts-2* seeds were treated with EMS (ethyl methylsulfonate), and the M2 plants were screened under 16 °C growth conditions. We acquired six plants (suppressors) that could rescue the chilling-sensitive phenotype of *bts-2*. Among these suppressors, a mutant *bts-r* (the suppressor of *bts-2*) grew relatively normal as wild-type plants at both 22 °C and 4 °C (Fig. 2a). The 3,3'-diaminobenzidine (DAB) and trypan blue staining confirmed that *bts-r* did not accumulate $H_2O_2$, and the extensive cell death did not occur as did in *bts-2* plants (Fig. 2b). In addition, *bts-r* exhibited comparable electrolyte leakage and survival rates as Col-0 by chilling treatment but exhibited a slight decrease of chlorophyll concentration than Col-0 under standard growth conditions (Supplementary Fig. 2a–d).

We then examined whether the *bts-r* mutant could resist iron deficiency. Under Fe-deficient conditions, unlike *bts-2*, the *bts-r* developed shorter roots, leaf chlorosis, and lower chlorophyll concentration, which were comparable to Col-0 (Fig. 2c–f), indicating that *bts-r* was not able to resist iron deficiency. Iron concentration assays further revealed that the *bts-r* mutants did not have the extraordinarily high iron concentration as did in the *bts-2* plants (Fig. 2g). These results indicated that the phenotypes of *bts-2* under the conditions of iron deficiency and chilling stress were largely rescued by *bts-r*.

## The suppressor gene encodes an RNA helicase RH24

To clarify the genetic background of *bts-r*, we crossed *bts-r* with *bts-2* to generate a segregating population. The F1 plants showed the chlorotic and stunted growth defect and temperature-sensitive phenotype at 16 °C. Among the F2 progeny, the temperature-sensitive mutant segregated in a 3:1 ratio (dwarf: wild type, 65/227; $\chi^2 = 1.245$; $p < 0.05$), indicating the trait that suppressing the phenotype of *bts-2* in *bts-r* was controlled by a single recessive locus. To identify the causative mutation in *bts-r*, the *bts-r* mutant was outcrossed to ecotype Landsberg *erecta* (L*er*). The F2 mutation was screened at 16 °C, and plants exhibiting a normal growth phenotype were selected for the map-based cloning. The *bts-r* mutant was mapped to the terminal of chromosome 2 between markers T3F17 and F11L15 (Fig. 3a). In parallel, the F2 progeny from *bts-r* backcrossed to *bts-2*, exhibiting *bts-r*-like phenotype and *bts-2*-like phenotype plants, were selected for whole-genome sequencing. Using MutMap, we narrowed down the *BTS-R* locus in the region between 18.7 Mb and 19.7 Mb on chromosome 2

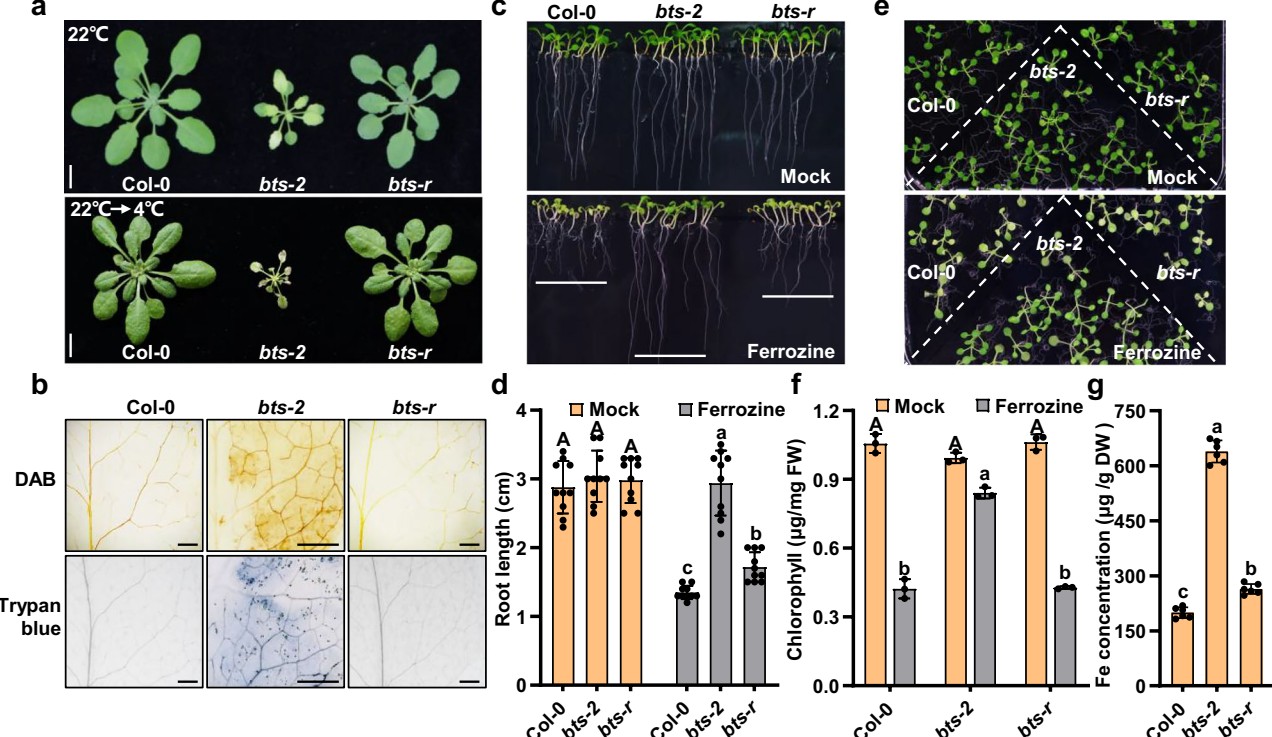

**Fig. 2 | *bts-r* (the suppressor of *bts-2*) exhibited reduced low-temperature sensitivity and iron deficiency tolerance. a** *bts-r* is not sensitive to low temperatures. Col-0, *bts-2*, and *bts-r* plants were grown at 22 °C for 4 weeks (upper panel) or grown at 22 °C for 4 weeks then transferred to 4 °C for 7 days (lower panel). Bar = 1 cm. **b** *bts-r* did not accumulate $H_2O_2$ under low temperature. The plants were grown at 22 °C for 4 weeks and transferred to 4 °C for another 2 days, and then the leaves were stained with DAB and trypan blue. Bar = 100 μm. The experiment was repeated at least three times with six individual leaves each time. **c** *bts-r* was sensitive to iron deficiency. The mutants were grown on ½ MS medium, which contains 100 μM Fe(II)-EDTA (hereafter as a mock), and the iron-deficient condition was made by applying 300 μM Ferrozine in ½ MS medium to chelate iron (lower panel) for 7 days. **d** Statistical analysis of root length for the plants in (**c**). The 7-day-old root length of the plants was measured. Data presented as means ± SD (*n* = 10 biological

replicates) by two-factor ANOVA with Tukey's HSD test (*P* ≤ 0.05). **e** The phenotypes of *bts-r* plants under iron deficiency. The plants were grown on ½ MS at 22 °C for 7 days and then transferred to ½ MS medium containing 100 μM Fe(II)-EDTA (mock) or 300 μM Ferrozine (iron deficiency) for 5 days, respectively. **f** Chlorophyll concentration of the plants in (**e**). Six plants per replicate were collected and used for chlorophyll concentration measurement. Data presented as means ± SD (*n* = 3 biological replicates) by two-factor ANOVA with Tukey's HSD test (*P* ≤ 0.05). **g** Iron concentration in Col-0, *bts-2*, and *bts-r* plants. Fe concentration is on the dry weight basis, and leaves from plants grown in soil under 22 °C for 4 weeks were used for Fe measurement. About twenty plants for per replicate were used. Data presented as means ± SD (*n* = 6 biological replicates) by one-way ANOVA with Tukey's HSD test (*P* ≤ 0.05). Source data are provided as a Source Data file.

(Fig. 3b). Four SNPs were identified in the candidate region, and only a single C-to-T nucleotide substitution in the second exon of *At2g47330* was identified in *bts-r*. This single mutation caused a premature stop codon in the gene at arginine 331 (Fig. 3c and Supplementary Fig. 3a). *At2g47330* gene encodes a putative DEAD (Asp-Glu-Ala-Asp) box RNA helicase (RH24), which contains a conserved DEAD-box and a helicase domain (Fig. 3d), and to our knowledge, its biological function has not been reported yet.

To confirm whether the *bts-r* phenotype was caused by the mutation in *RH24*, a wild-type *RH24* gene under the control of its own promoter was transformed into *bts-r*. It has been reported that the DEAD-box domain was essential for RNA helicase enzymatic activity in planta[23]. Thus, we also generated *pRH24:RH24^DAAD* transgenic lines harboring the DAAD mutation version of RH24 in the *bts-r* mutant. The result showed that two independent T2 transgenic plants fully complemented the temperature-sensitive and iron deficiency-tolerance phenotypes of *bts-2* (Fig. 3e and Supplementary Fig. 3b), and the *pRH24:RH24^DAAD* transgenic plants phenocopy the *bts-2* mutant, suggesting the DAAD domain is required for the RH24 function. In addition, we also isolated two T-DNA insertion lines of At2g47330, Salk_144439 and Salk_045730 (TAIR), and we named them as *rh24-1* and *rh24-2*, respectively (Fig. 3c and Supplementary Fig. 3c). The *rh24-1/bts-2* double mutant was generated by crossing *bts-2* with *rh24-1*. The

deletion of *RH24* completely abolished the phenotypes of *bts-2*, and RH24, but not RH24^DAAD, could recapitulate the phenotypes (Fig. 3e and Supplementary Fig. 3b). These findings indicate that the phenotype of *bts-r* was indeed caused by a null mutation in *RH24*.

A previous study has demonstrated that the RH24 homolog AtRH7 plays a vital role in pre-rRNA processing and cold tolerance in Arabidopsis[15]. The amino acid sequence alignment showed that there was a relatively high similarity between RH24 and RH7 (Supplementary Fig. 3d). However, the loss of *RH7* could not suppress the chilling sensitivity of *bts-2* (Supplementary Fig. 3e, f), suggesting that RH24 specifically suppresses the chlorotic and stunted phenotype of *bts-2*.

## RH24 is an ATP-dependent RNA helicase

Apparently, the RNA helicase activity seemed critical for RH24 to suppress the phenotypes of *bts-2* (Fig. 3). We then ask if RH24 is a bona fide RNA helicase. RNA helicases usually acquire energy by hydrolyzing ATP to unwind the RNA duplex, which is involved in all aspects of RNA metabolism[19]. Therefore, to examine the biochemical functions of RH24, we purified recombinant RH24 protein from *Escherichia coli* (Supplementary Fig. 4a) and tested its RNA helicase activity by measuring its unwinding activity and ATPase activity.

We first determined the unwinding activity of RH24 using molecular beacon helicase assays (MBHA) as described previously[43–45]. The

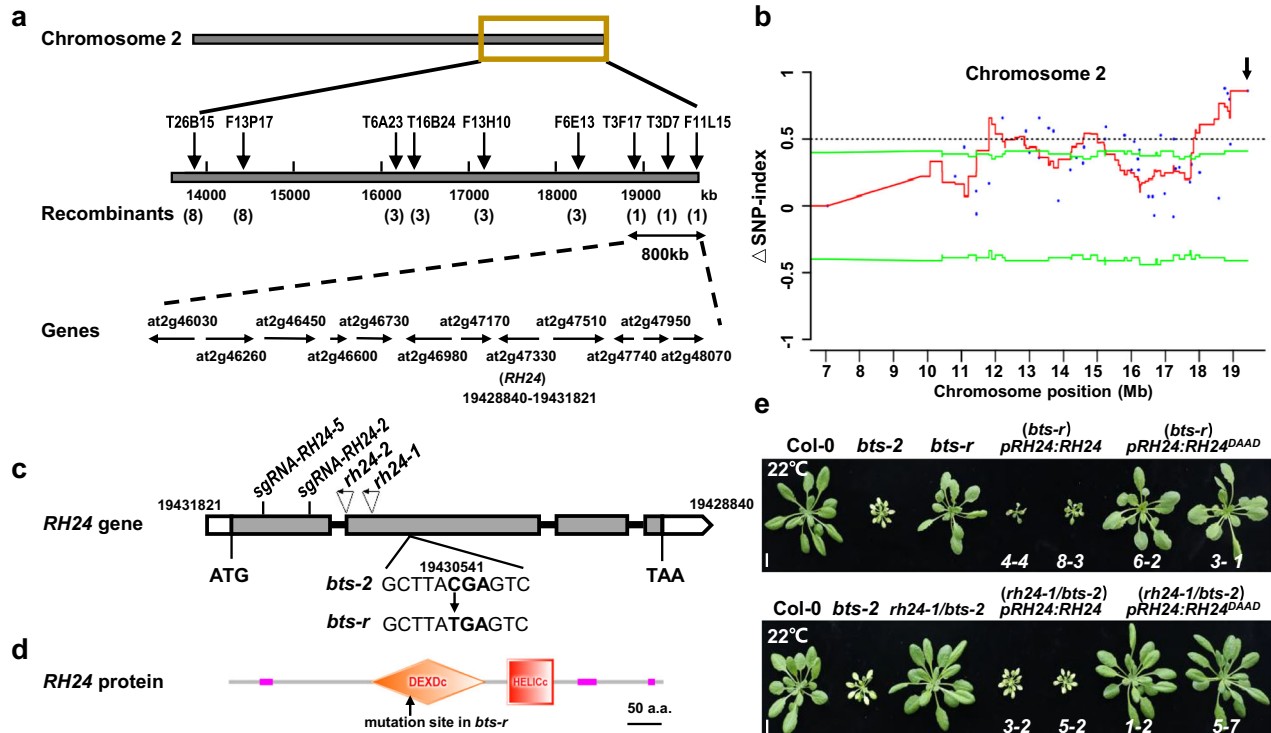

**Fig. 3 | Map-based cloning of *BTS-R*. a** Genetic map of the *BTS-R* locus region in Arabidopsis. Positions of the markers used for mapping are indicated. The number of recombinants is indicated in parentheses. Predicted genes are shown by arrows. **b** The distribution of ΔSNP-index on chromosome 2. The MutMap analysis shows a genomic region with the highest ΔSNP-index peak harboring the candidate mutation. ΔSNP-index is obtained by subtracting the SNP-index value of *bts-2* bulk from that of *bts-r* bulk. Blue dots corresponding to the ΔSNP-index value of homozygous SNPs were identified between *bts-r* and *bts-2* mutant genomes. The red line represents the sliding window average values of ΔSNP-index indices of 1 Mb interval with a 1 kb increment. The green line represents the screening threshold value (95% confidence interval). The confidence interval was calculated by the bootstrap method to determine whether ΔSNP-index was significant. The dashed line represents ΔSNP-index = 0.5. **c** The genomic structure of the *BTS-R* gene. White and gray boxes indicate untranslated regions (UTR) and exons, and lines between boxes indicate introns. The positions of two sgRNA:Cas9 targets of RH24, T-DNA insertions, and the nucleotide substitutions in *bts-r* are shown. **d** The protein structure of RH24. RH24 protein contains a DEAD-box and a helicase domain. The early stop codon generated by the C to T mutation in *bts-r* is shown by the arrow. **e** Genetic complementation of *bts-r* by *RH24*. Phenotypes of Col-0, *bts-2*, *bts-r*, *pRH24:RH24* (*bts-r*), *pRH24:RH24*^DAAD^ (*bts-r*), *rh24-1/bts-2*, *pRH24:RH24* (*rh24-1/bts-2*), and *pRH24:RH24*^DAAD^ (*rh24-1/bts-2*) are shown. The genes were driven by the native promoter of *RH24*. The plants were grown at 22 °C for 4 weeks. Bar = 1 cm.

MBHA monitored the activity of the helicase to release the molecular beacon that binds to a complementary strand (Fig. 4a). In the presence of recombinant maltose-binding protein (MBP)-RH24, the fluorescence of double-stranded RNA (dsRNA) was significantly decreased, but MBP-RH24^DAAD^ and MBP did not affect the fluorescence (Fig. 4b). Meanwhile, we confirmed the unwinding activity of RH24 using recombinant His-RH24 and biotin-labeled partial RNA duplexes in a strand displacement assay[46,47], where the dsRNA was gradually unwound following the increment of ATP (Fig. 4c). However, no unwinding activity was observed for His-RH24^DAAD^ (Fig. 4d). Furthermore, we investigated the ATPase activity of RH24 using the pyruvate kinase/lactate dehydrogenase coupled enzyme assays[23], where we found that His-RH24 has ATPase activity, but His-RH24^DAAD^ does not possess the activity (Fig. 4e). All the above results demonstrated that RH24 is an ATP-dependent RNA helicase, and the DEAD-box domain is essential for its RNA helicase activity. Consistent with the phenotype of *pRH24:RH24*^DAAD^ (*bts-r*) transgenic plants (Fig. 3e), we conclude that, most probably, the dsRNA unwinding activity of RH24 is indispensable for suppressing the chilling sensitivity of *bts-2*.

To further explore the role of RH24 in plant chilling tolerance, we generated the transgenic *RH24-ox* plants by overexpressing *RH24-Flag*. Then, we tested whether chilling stress influences the protein levels of RH24 under cycloheximide (CHX) treatment. Surprisingly, RH24 protein levels were remarkably elevated at lower temperature (4 °C) but repressed at higher temperature (26 °C), and CHX treatments did not

alter the trend (Fig. 4f and Supplementary Fig. 4b), indicating that RH24 is accumulated at a post-transcription level under chilling stress. It also supports the observation in the reversion of phenotypes of *bts-2* at 26 °C due to the reduction of RH24 protein at high temperatures.

## RH24 regulates plant chilling and iron deficiency responses

To elucidate the biological function of RH24, we determined the effect of RH24 on plant sensitivity to chilling stress by assessing their hypocotyl elongation in the dark based on Liu et al.[48]. The hypocotyl elongation of *rh24* null mutants was higher than *bts-2* and Col-0 at 4 °C but not at 22 °C (Supplementary Fig. 5a), and the hypocotyl elongation of *35S:RH24-ox* transgenic plants were significantly reduced when compared with Col-0, especially under the iron oversupply (300 μM) condition (Fig. 5a, b), indicating that RH24 negatively regulated the plant chilling tolerance (4 °C). We then analyzed the iron deficiency resistance in *rh24* mutants. The *rh24-1* and *rh24-2* mutants exhibited increased sensitivity to iron deficiency, revealed by the reduced root length (Supplementary Fig. 5b). The *rh24-1* and *rh24-2* mutant also exhibited enhanced tolerance to iron oversupply (300 μM) under chilling conditions, which exhibited higher chlorophyll concentration than that of Col-0 (Fig. 5c, d). Moreover, the iron concentration was lower in *rh24* mutants than that of wild type and *bts-2*, and the loss of RH24 reduced the iron concentration in *bts-2* (Fig. 5e). In addition, we measured iron accumulation in leaves of *rh24-1*, *rh24-2*, and two RH24-ox lines by Perls Fe staining. The result showed that *rh24-1* and *rh24-2*

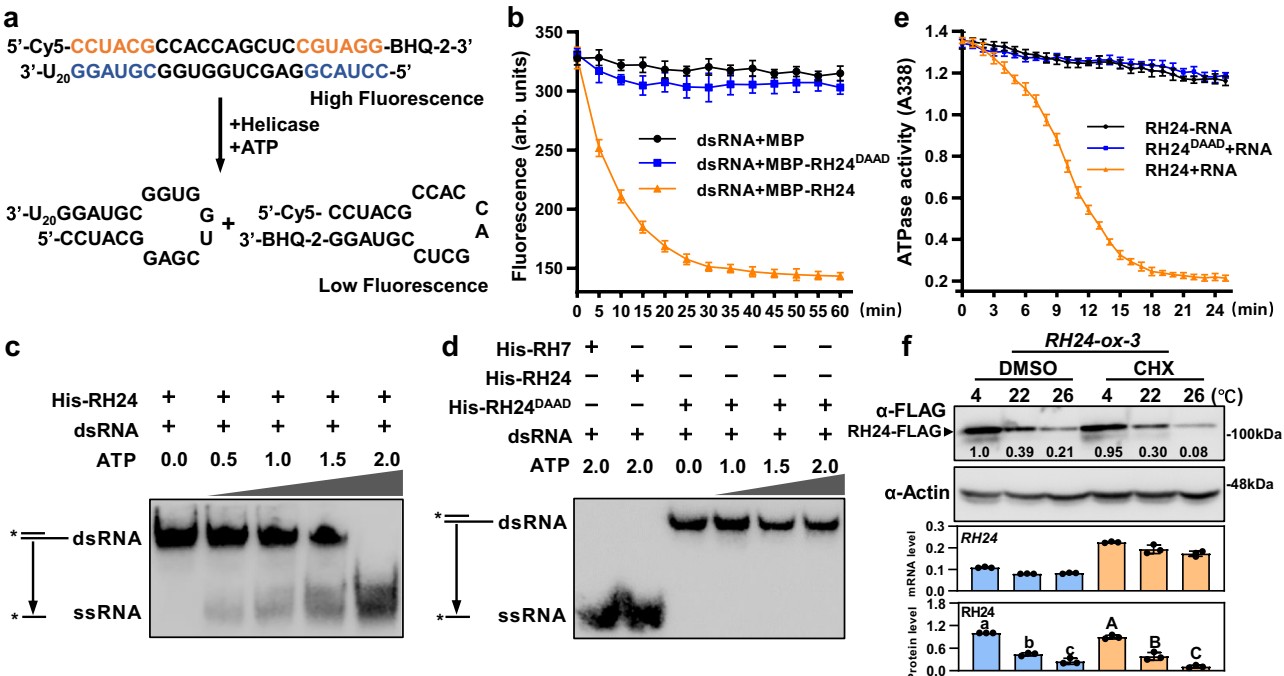

**Fig. 4 | RH24 is an ATP-dependent RNA helicase and accumulates under low temperatures. a** The molecular beacon helicase assays (MBHA). RH24 helicase unwinds a molecular beacon from a longer oligonucleotide. The fluorescent strand in the 3′-U20 overhanging RNA substrate was modified with Cy5 at the 5′ end and BHQ-2 at the 3′ end. **b** Helicase activity of RH24 in MBHA. The decreasing intensity of the fluorescence indicates the activity of the helicase in the MBAH assays. Data are presented as mean values ± SD. **c** RH24 has ATP-dependent dsRNA unwinding activity. The unwinding activity was reflected by the double-stranded RNA template converting to a single-stranded product. The asterisk on the depiction of the short strand of the RNA duplex represents the biotin-labeled RNA. (ds: double strand, ss: single strand). **d** RH24DAAD does not have the dsRNA unwinding activity. The assay was the same as in (**c**). The DEAD-box RNA helicase RH7 and RH24 were used as the positive controls. **e** ATPase activities of His-RH24 and His-RH24DAAD. The ATPase

activities of His-RH24 and His-RH24DAAD were measured in the presence or absence of plant total RNA. Data are presented as mean values ± SD. **f** RH24 protein was accumulated under chilling stress. The 35S:RH24-ox Line 3 was grown on ½ MS at 22 °C for 12 days and then treated with 2 μM CHX at 4, 22, and 26 °C for 2 days, respectively. The protein levels of RH24 were examined by immunoblotting analysis. The expression levels of RH24 were examined by RT-qPCR. The band abundance was quantified by Image J. The relative protein level of RH24 was quantitatively analyzed with three repeats (lower panel). Different letters indicate the significant differences based on a two-factor ANOVA with Tukey's HSD test ($P ≤ 0.05$, Data are means ± SD, $n = 3$ biological repeats). CHX: cycloheximide. The above experiments were repeated three times with similar results. Source data are provided as a Source Data file.

mutant leaves accumulated less iron than Col-0 and RH24-ox lines at 4 °C (Supplementary Fig. 5c). Under normal growth condition, although *35S:RH24-ox* transgenic plants did not exhibit significant difference in chlorophyll concentration and root length under chilling stress or iron deficiency condition when compared with Col-0 (Supplementary Fig. 5d, e), the *35S:RH24-ox* transgenic plants displayed increased chilling sensitivity by iron oversupply (Fig. 5c, d). Nevertheless, these results indicate that RH24 regulates plant chilling and iron deficiency responses.

To further confirm the role of *RH24* in the chilling sensitivity of *bts-2*, we also generated the *RH24* knockout transgenic lines in *bts-2* using CRISPR/Cas9 techniques (Supplementary Fig. 5f, g) and examined the iron deficiency resistance in those double mutants. When grown on an iron-deficient medium, both *Cas9-RH24/bts-2* and *rh24-1/bts-2* double mutants were much more sensitive to the iron deficiency than *bts-2*, with significantly shorter roots, although they were slightly higher than Col-0 (Fig. 5f, g). In addition, *Cas9-RH24/bts-2* plants developed normally as Col-0 at 22 °C and 4 °C (Supplementary Fig. 5h, i). Based on these findings, we conclude that RH24 regulates plant chilling and iron deficiency responses, and depletion of RH24 rescues the defect phenotypes of *bts-2*. In addition, these data reveal that RH24 is genetically epistatic to BTS.

## ILR3 accumulation largely depends on RH24

Since RH24 is genetically epistatic to BTS, we want to know how RH24 biochemically regulates plant chilling and iron deficiency responses.

We first attempted to understand the relationship between RH24 and BTS. We performed the bimolecular fluorescence complementation (BiFC) assays in *Nicotiana benthamiana* and *Arabidopsis* protoplast to examine if there was an interaction between RH24 and BTS. The result showed that RH24 interacted with BTS in the nucleus (Fig. 6a); however, no interaction was observed between RH24DAAD and BTS (Fig. 6a). We also performed the split-luciferase complementation (split-LUC) imaging assays and co-immunoprecipitation (co-IP) assays. The results confirmed the interaction of BTS and RH24 but not BTS and RH24DAAD (Supplementary Fig. 6a, b). These results suggest that the helicase activity of RH24 is likely required for the interaction of RH24 and BTS.

To investigate the biological significance of the RH24 and BTS interaction, we constructed and expressed the 5′UTR-CDS-3′UTRs of *BTS* in Arabidopsis protoplast, and then examined whether RH24 affects the protein levels of BTS. However, we found neither RH24 nor RH24DAAD essentially influenced BTS protein accumulation (Fig. 6b). These data suggest that BTS is unlikely the direct target of RH24. It has been reported that the BTS-interacting protein ILR3 is a suppressor of BTS and null mutation of ILR3 could completely suppress *bts-2* phenotypes[49,50]. We, therefore, were interested to know if RH24 could positively regulate ILR3.

To examine if there was an interaction between RH24 and ILR3, we performed the BiFC assays, split-LUC assays, and co-IP assays. The results confirmed that RH24 interacted with ILR3 and were co-localized in the nucleus (Fig. 6c and Supplementary Fig. 6c–e). We then performed the RT-qPCR assays for the tissue-specific expression

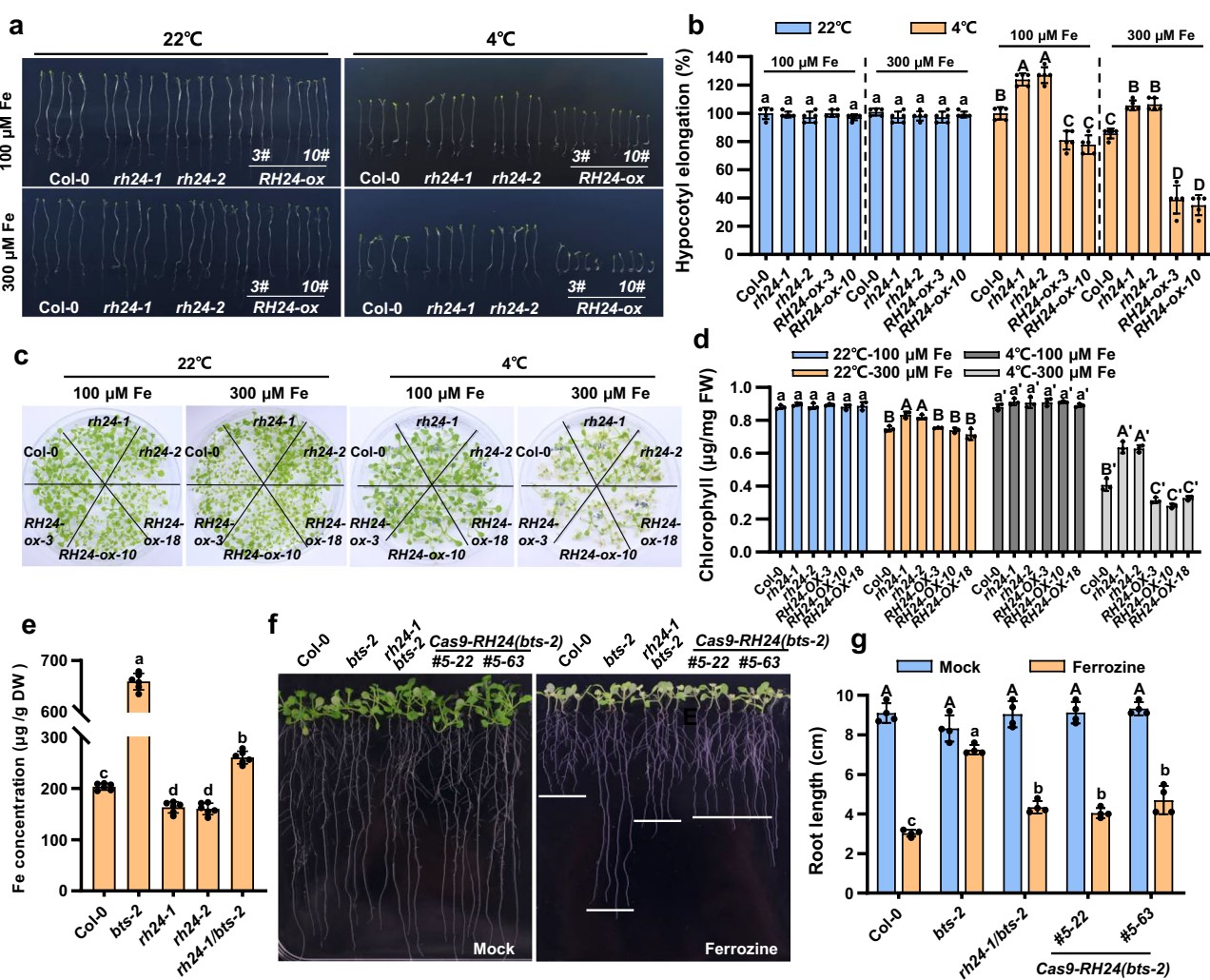

**Fig. 5 | RH24 is genetically epistatic to BTS. a** Chilling sensitivity of Col-0, *rh24-1*, *rh24-2*, and *RH24-ox* plants in darkness. The plants were grown on ½ MS medium, which contains 100 or 300 µM Fe(II)-EDTA. Left panel, 22 °C for 10 d; right panel, 4°C in the dark for 21 d. **b** Quantification of hypocotyl elongation of plants shown in (**a**). Hypocotyl elongation of the plants were compared with Col-0 at 22 °C or 4 °C under control conditions (100 µM Fe). The value of Col-0 was set as 100%. Data presented as mean ± SD (*n* = 5 biological replicates) by two-factor ANOVA with Tukey's HSD test (*P* ≤ 0.05). **c** The phenotype of Col-0, *rh24-1*, *rh24-2*, and *RH24-ox* plants under iron and chilling stresses. Plants were grown on ½ MS medium at 22 °C for 7 days and then transferred to ½ MS medium which contains 100 or 300 µM Fe(II)-EDTA at 22 °C for 1 week (left panel) and then treated with 4 °C for another 2 weeks (right panel). **d** Chlorophyll concentration of the plants in (**c**). Six plants of replicate were collected and used for chlorophyll concentration measurement.

Data presented as means ± SD (*n* = 3 biological replicates) by two-factor ANOVA with Tukey's HSD test (*P* ≤ 0.05). **e** Iron concentration in Col-0, *bts-2*, *rh24-1*, *rh24-2*, and *rh24-1/bts-2* plants. The plants were grown under 22°C for 4 weeks, and the leaves were used for Fe measurement. About twenty plants were used for each replicate. Data presented as means ± SD (*n* = 6 biological replicates) by one-way ANOVA with Tukey's HSD test (*P* ≤ 0.05). **f** The phenotype of *rh24-1/bts-2* mutants under iron deficiency. The mutants were grown on ½ MS for 5 days and then transferred to ½ MS medium, which contains 100 µM Fe(II)-EDTA (mock) or 300 µM Ferrozine (iron deficiency) in the medium (right panel) for 7 days at 22 °C. **g** Statistical analysis of root length for the plants in (**f**). The root length of 12-day-old plants was measured. Data presented as means ± SD (*n* = 4 biological replicates) by two-factor ANOVA with Tukey's HSD test (*P* ≤ 0.05). Source data are provided as a Source Data file.

pattern of *RH24*, *BTS*, and *ILR3*. The results showed that the *RH24*, *BTS*, and *ILR3* were all highly expressed in young leaves (Supplementary Fig. 6f). We also expressed the 5´UTR-CDS-3´UTRs of *ILR3* in Arabidopsis protoplast, and then examined whether RH24 affects the protein levels of ILR3. The result showed that RH24 increased the ILR3 protein levels (Fig. 6d). Moreover, the presence of protease inhibitor MG132, did not further increase the accumulation of ILR3, suggesting that ILR3 protein is stable in the presence of RH24 (Fig. 6d). To determine whether RH24 potentially unwinds *ILR3* mRNA in planta, we performed an RNA immunoprecipitation (RIP)-qRCR assay in *35S:RH24-Flag* transgenic plants. Notably, there was a substantial enrichment for *ILR3* in RH24 RIP relative to IgG control, indicating that RH24 interacts with *ILR3* mRNA (Fig. 6e). Since RH24 did not influence

transcription levels of *ILR3* (Supplementary Fig. 6g), we hypothesize that RH24 unwound *ILR3* mRNA in vivo. We next generated the *ILR3-ox/rh24-1* plants by crossing *rh24-1* with *ILR3-ox* transgenic plants to investigate the effects of RH24 on ILR3 accumulation in planta. Surprisingly, the accumulation of ILR3 was largely dependent on RH24 in the transgenic plants, and chilling stress led to ILR3 accumulation in Col-0 background, but the accumulation was almost completely abolished in the *rh24-1* mutant (Fig. 6f and Supplementary Fig. 6h). In addition, in absence of RH24, the protease inhibitor MG132 did not revert the accumulation of ILR3 in *ILR3-ox/rh24-1* plants, suggesting that ILR3 protein levels were dependent on RH24 and RH24 seems to stabilize ILR3 (Fig. 6f and Supplementary Fig. 6h). It is worth noting that temperatures had no significant effect on *ILR3* transcription,

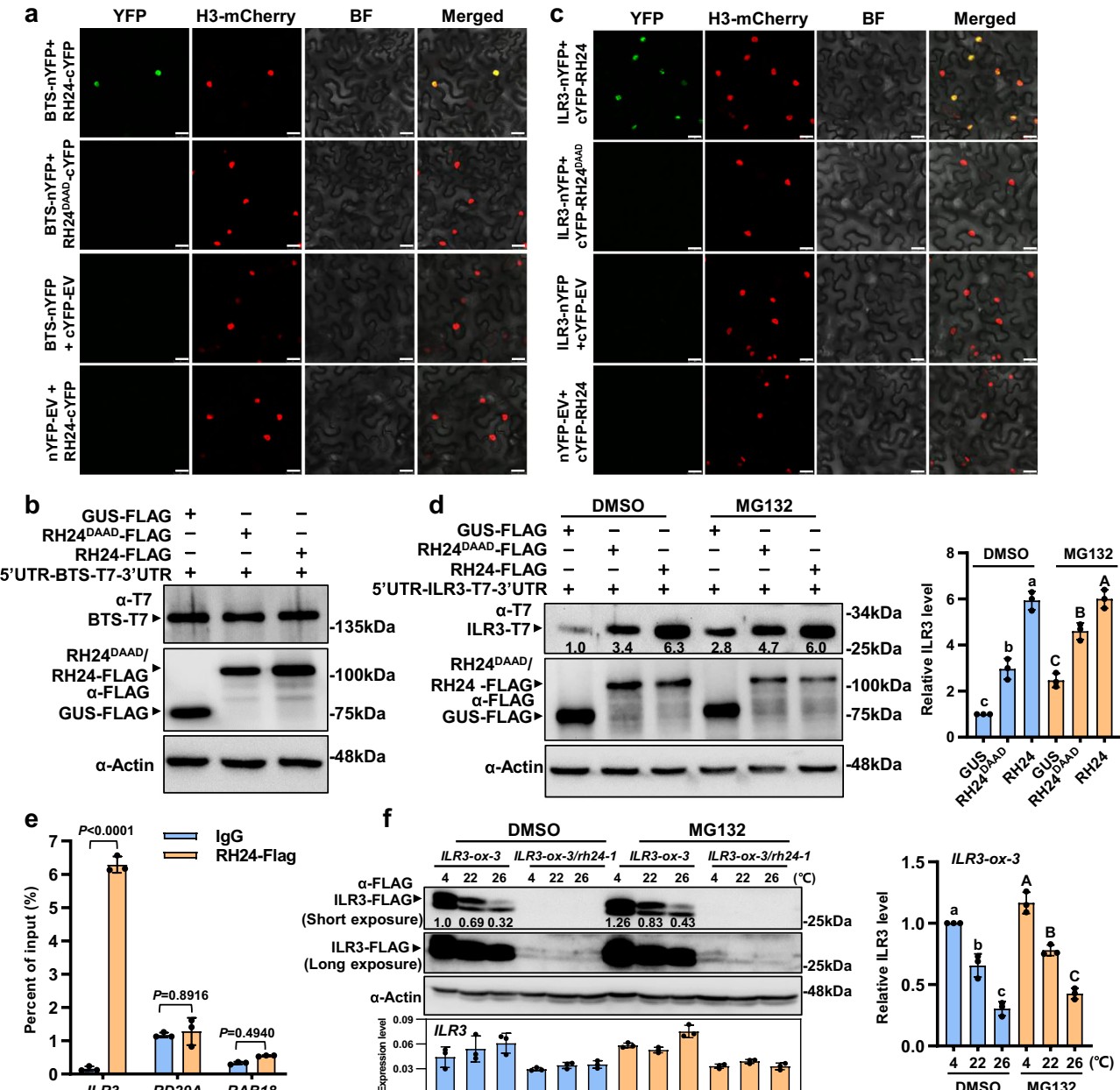

**Fig. 6 | ILR3 protein accumulation depends on RH24. a** RH24 interacts with BTS in BiFC assays. The proteins were transiently expressed in *N. benthamiana* leaves. The plants stably expressing Histone3-mcherry served as a nucleus marker. BTS-nYFP and cYFP-EV or nYFP-EV and RH24-cYFP were used as a negative control. YFP fluorescence was assigned with pseudocolor green. Bar = 25 μm. BF: bright field. **b** RH24 did not affect the protein levels of BTS. *35S:5'UTR-BTS-T7-3'UTR* and *35S:RH24-FLAG* or *35S:RH24^DAAD-FLAG* were expressed in Arabidopsis protoplasts. The protein levels were subjected to immunoblotting assays at 16 h after transformation. **c** RH24 interacts with ILR3 by BiFC assays. Bar = 25 μm. **d** ILR3 accumulates in the presence of RH24. *35S:5'UTR-ILR3-T7-3'UTR* and *35S:RH24-FLAG* or *35S:RH24^DAAD-FLAG* were expressed in Arabidopsis protoplasts. 50 μM MG132 was added to the samples at 12 h after transfection. After an additional 6 h incubation, the protoplasts were subjected to immunoblot assays. The band abundance was quantified by Image J. The relative protein level of ILR3 was quantitatively analyzed

for three repeats (right panel). Data presented as means ± SD (*n* = 3 biological replicates) by two-factor ANOVA with Tukey's HSD test (*P* ≤ 0.05). **e** RIP-qPCR assays for RH24 and *ILR3* mRNA. RIP-qPCR assays were performed using *35S:RH24-FLAG* plants. The plants were grown on ½ MS for 14 days. RT-qPCR was used to evaluate the precipitated mRNA. Values are the mean of the percentage of input in three independent experiments. Data presented as means ± SD (*n* = 3 biological replicates) by two-factor ANOVA with Tukey's HSD test (*P* ≤ 0.0001). **f** Accumulation of ILR3 protein under chilling stress is dependent on RH24. The transgenic plants *ILR3-ox-3* and *ILR3-ox-3/rh24-1* were grown on ½ MS at 22 °C for 12 days and then treated with 2.5 μM MG132 at 4, 22, or 26 °C for 2 days, respectively. The expression levels of *ILR3* were examined by RT-qPCR. The relative protein level of ILR3 was quantitatively analyzed for three repeats (right panel). Data presented as means ± SD (*n* = 3 biological replicates) by two-factor ANOVA with Tukey's HSD test (*P* ≤ 0.05). Source data are provided as a Source Data file.

suggesting that low temperatures regulate ILR3 accumulation at post-transcription level.

 We then ask if RH24 unwinds ILR3 mRNA to increase the protein levels. To explore the mechanism, we analyzed the transcriptome of

*rh24* and RH24-ox lines by the integrated genome (IGV) browser. We observed that *ILR3* displayed mis-spliced transcripts in *rh24* mutants (Supplementary Fig. 6i), and it was also seen in RH24 homologies AtRCF1/AtRH42, OsRH42, and PRP5 that were involved in pre-mRNA

splicing[23,24,51,52], suggesting that RH24 acts similarly as its homologies. The RT-PCR result showed that the levels of intron-containing isoform of *ILR3* were abundant in *rh24* mutants but were abolished in the *RH24-ox* lines (Supplementary Fig. 6j), reinforcing the conclusion that RH24 plays a potential role in pre-mRNA splicing.

## RH24 and ILR3 are in phase-separated condensates

In fact, RH24 interacts with ILR3 in the plant nuclear bodies (Fig. 6c), and ILR3 alone was able to locate in the nuclear bodies in the presence of RH24 under 8 °C, but not in *rh24* mutant where no nuclear bodies could be observed (Supplementary Fig. 6k), revealing that RH24 facilitates ILR3 nuclear body formation. We then transiently co-expressed BTS-mCherry, cYFP-RH24, and ILR3-nYFP fusion proteins to check whether BTS was co-localized with RH24 and ILR3 in the cells. Notably, RH24 and ILR3 were co-localized with BTS (Fig. 7a). Furthermore, we found that RH24-ILR3 nucleus bodies were the phase-separated condensates by analyzing the liquid-like properties using fluorescence recovery after photobleaching (FRAP) approach. We transiently expressed cYFP-RH24 and ILR3-nYFP in *N. benthamiana* leaf epidermal cells. After photobleaching, approximately 58.6% of the RH24-ILR3 nucleus bodies signals were gradually recovered over time (Fig. 7b and Supplementary Fig. 7a), indicating a rapid redistribution of these RH24-ILR3 nucleus bodies from the nucleoplasm. In addition, fusion of RH24-ILR3 nucleus bodies were also observed in *N. benthamiana* cells (Fig. 7b). These results reveal that ILR3 and RH24 are in phase-separated condensates. We also performed a cold treatment (4 °C) on the plants before observing the fluorescence signals. After cold treatment for 1 hour, the condensates were significantly increased in *N. benthamiana* cells and *Arabidopsis* protoplasts (Fig. 7c and Supplementary Fig. 7b, c). Meanwhile, we observed that BTS further facilitated the condensate formation (Fig. 7c and Supplementary Fig. 7c). These results suggest that the RH24-ILR3 phase-separated condensates were regulated by cold stress and BTS.

## ILR3 is genetically associated with RH24 in iron and chilling stress responses

As ILR3 accumulation is dependent on RH24, we then examined the chilling response of the loss-of-function *ilr3-2* mutant. The *ilr3-2* mutants grew comparably well as Col-0 at 4 °C and 8 °C, but they showed visible chlorosis on the leaves at 22 °C and 26 °C, and these phenomena were not seen in Col-0 (Supplementary Fig. 7d). These results showed that the *ilr3* loss-of-function mutant displayed an opposite phenotype of *bts-2* mutant. Consistently, the *ILR3-ox/rh24-1* plants exhibited enhanced chilling tolerance and elevated chlorophyll concentration when compared with *ILR3-ox* (Fig. 7d). Meanwhile, *ILR3-ox/rh24-1* plants also showed reduced iron concentration, increased sensitivity to Fe deficiency, and enhanced tolerance to the excessive amount of iron when compared with *ILR3-ox* (Fig. 7e and Supplementary Fig. 7e–h), indicating that RH24 is required for the chilling sensitivity of *ILR3-ox* plant.

Previously, we reported that the accumulation of iron facilitates pathogen proliferation in plants[41]. We, therefore, were interested in investigating how *rh24* responded to pathogen infection. The bacteria growth curve assays showed that *rh24-1* and *ILR3-ox/rh24-1* plants displayed significantly enhanced resistance to *Pseudomonas syringae* pv. *tomato* (*Pst*) DC3000 infection when compared with Col-0, *bts-2*, and *ILR3-ox* plants (Supplementary Fig. 7i). These results collectively support the conclusion that RH24 unwinds *ILR3* mRNA and promotes the accumulation of ILR3, thereby facilitating the chilling sensitivity of *bts-2* and plant immune response.

## Discussion

Chilling stress is one of the most common stresses in nature. It affects plant growth and development, disrupts cellular homeostasis, and impairs photosynthesis and enzyme activities in plants[53]. Many environmental factors can deliberate chilling stress, such as light, phytohormones, circadian clock, and nutrients[1,54]. Iron serves as a redox catalytic factor for many key enzymes, being the most important mineral nutrient in plants[29]. Iron homeostasis is essential for plant growth and has been found critical for plants to alleviate salinity, drought, or heavy metal stresses[55,56]. Iron scarcity increases plant sensitivity to salinity or drought stresses[57,58]. However, iron overload damages plants by facilitating the production of highly active ROS[55]. Despite the vital roles of iron in many stress responses, little is known whether iron homeostasis is involved in the response to chilling stress.

Plants have evolved sophisticated mechanisms to precisely regulate iron homeostasis. A series of bHLH transcription factors directly regulate the response of iron deficiency in Arabidopsis[59,60]. Among these iron deficiency-responsive transcription factors, ILR3 plays a central role in regulating iron homeostasis. When iron supply is sufficient, BTS destabilizes ILR3 and fine-tunes iron uptake to prevent plants from being poisoned by iron overload[38]. Notably, *ilr3* loss-of-function mutant is insensitive to chilling stress, but the overexpression lines are sensitive to chilling stress (Fig. 7d and Supplementary Fig. 7d). This phenomenon is opposite to the phenotype of *bts-2*, in which it is sensitive to chilling stress (Fig. 1b and Supplementary Fig. 1a–d), implying that BTS/ILR3-mediated iron homeostasis is involved in chilling stress response.

One of the major damages caused by low temperatures is to affect the PSI function. It has been found that increasement of iron uptake is required for plant chilling tolerance, which is largely attributed to the increased iron load to PSI[37]. However, the endogenous iron may be maintained at certain levels, as we found that iron oversupply increased the plant chilling sensitivity (Supplementary Fig. 1h). Consistently, the *bts-2* and *ILR3-ox* plants, which both accumulate high levels of iron, exhibited strong chilling-sensitive phenotypes (Supplementary Fig. 1a and Fig. 7d). However, the transcriptome data and the endogenous iron concentration measurements revealed that low temperature induced iron transportation and storage in the plants (Fig. 1a), and iron was slightly accumulated under such circumstances (Fig. 1c). Therefore, certain iron accumulation is likely required for chilling tolerance. In fact, *ilr3-2*, a loss-of-function mutant, grew relatively healthier at a lower temperature (4 °C) than that at a higher temperature (26 °C) when compared to the respective Col-0 plants (Supplementary Fig. 7d). These data indicate that reducing iron levels, such as that in *ilr3-2* plants, increased their low-temperature tolerance.

BTS/ILR3-regulated iron homeostasis is critical for plant chilling tolerance, but how they linked to temperature fluctuations? Here, we discovered that removal of RNA helicase RH24 could fully suppress the chilling sensitivity of *bts-2* (Fig. 3e). RNA helicases are widely present in all eukaryotes and most of the prokaryotes, where they act as RNA chaperones to catalyze or unwind dsRNA[19]. It has been reported that many RNA helicases are involved in the regulation of cold stress in plants by processing specific RNAs[15,27]. In yeast, the DEAD-box RNA helicase Ded1p undergoes heat-induced phase separation, in which it regulates the translation of housekeeping and stress-induced mRNAs[61].

Intriguingly, *BTS* mRNA seems not to be the substrate of RH24, although RH24 and BTS interact physically, and *bts-2* chilling sensitivity is suppressed in the absence of RH24 (Fig. 3 and Supplementary Fig. 6). Instead, RH24 binds with and likely splices *ILR3* mRNA, which consequently regulates ILR3 protein levels although it also seems to stabilize ILR3 (Fig. 6d–f and Supplementary Fig. 6i, j). Several DEAD-box RNA helicases have been reported to modulate the pre-mRNA splicing fidelity control and mediate the ATP-dependent conformational change of small nuclear ribonucleoprotein particles (snRNPs). These snRNPs were required for spliceosome formation during pre-mRNA splicing process to ensure the accuracy and efficiency of gene expression[24,52,62]. Given the distinct levels of intron-containing isoform of *ILR3* in *rh24* mutants versus that in *RH24-ox* lines (Supplementary Fig. 6j), we hypothesize that RH24 binds to *ILR3* pre-mRNA to splice the

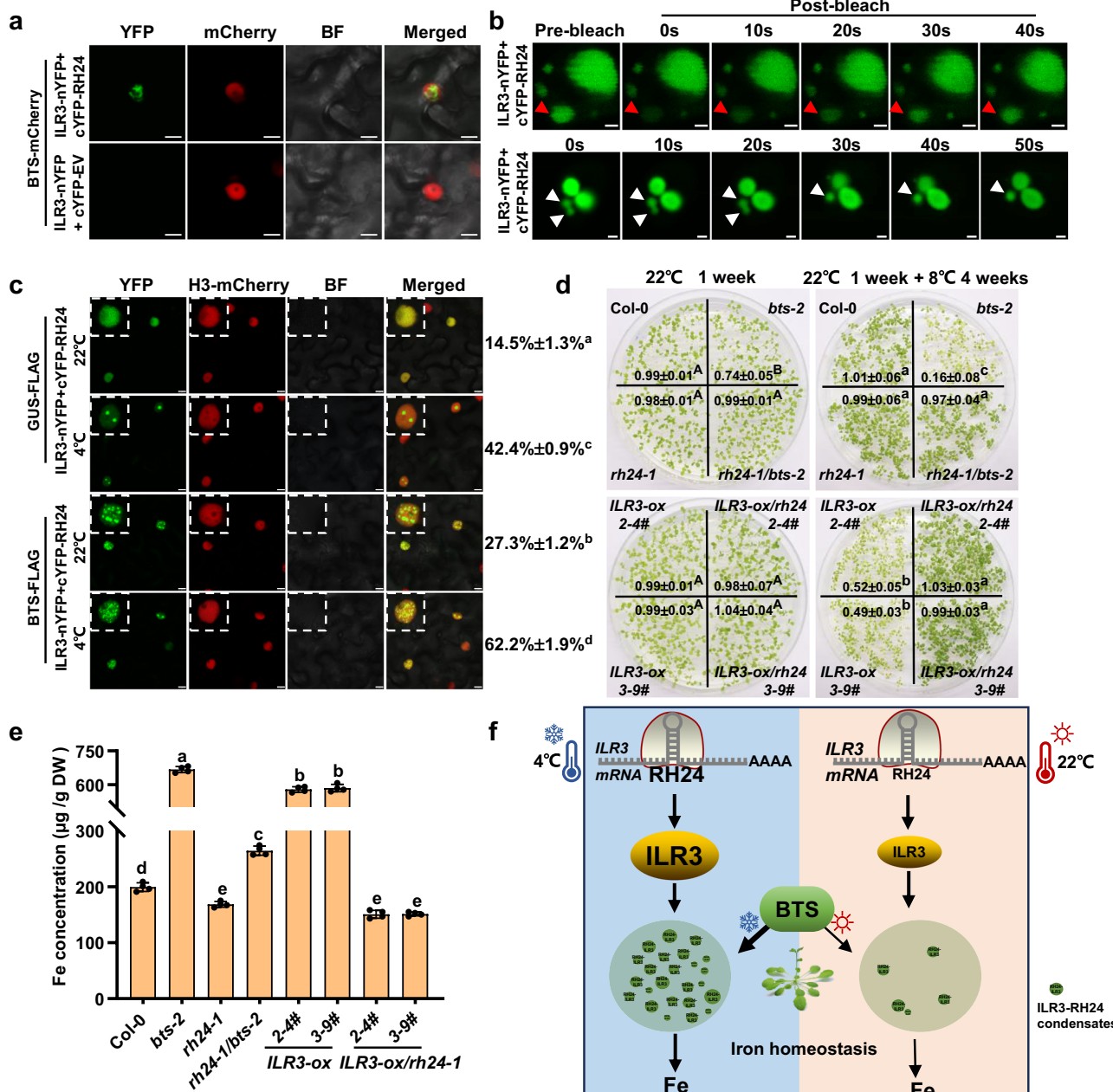

**Fig. 7 | ILR3 and RH24 are in condensates and BTS facilitates the condensate formation. a** RH24 interacts with ILR3 and were co-localized with BTS in the nucleus. The proteins were transiently expressed in *N. benthamiana* leaves. YFP fluorescence was assigned with pseudocolor green. Bar = 10 μm. **b** ILR3 and RH24 are in phase-separated condensates. The upper panel shows the FRAP of ILR3-nYFP and cYFP-RH24 that were transiently co-expressed in *N. benthamiana* leaf epidermal cells. Red arrows indicate the nuclear bodies that are bleached. The images at the bottom show the fusion of the nuclear bodies. White arrows indicate the nuclear bodies that undergo fusion. Bar = 2 μm. Images are representative of three independent experiments. **c** BTS and cold treatments facilitate condensate formation. ILR3-nYFP, cYFP-RH24, GUS-FLAG, and BTS-FLAG were transiently expressed in *N. benthamiana* leaves at 22 °C for 2 days and then transferred to 22 °C or 4 °C for another 1 h. Bar = 7.5 μm. The percentage of ILR3-RH24 condensations were presented. Images are representative of three independent experiments. **d** ILR3-mediated chilling sensitivity depends on RH24. *2-4#* and *3-9#* are two

independent *35S:ILR3-ox* transgenic lines. The plants were grown on ½ MS at 22 °C for 1 week or at 22 °C for 1 week and treated at 8 °C for another 4 weeks. The concentration of chlorophyll (μg/mg FW) was presented. Data presented as means ± SD (*n* = 3 biological replicates) by two-factor ANOVA with Tukey's HSD test (*P* ≤ 0.05). **e** Iron concentration in *ILR3-ox* and *ILR3-ox/rh24* plants. The leaves from plants grown in soil under 22 °C for 4 weeks were used for Fe measurement. Data presented as means ± SD (*n* = 4 biological replicates) by one-way ANOVA with Tukey's HSD test (*P* ≤ 0.05). **f** The working model: RH24 unwinds *ILR3* mRNA to facilitate the accumulation of ILR3. Under chilling stress, RH24 is accumulated, which sequesters ILR3 in phase-separated bodies by forming RH24-ILR3 condensates in the nucleus. BTS and cold stress facilitate RH24-ILR3 condensate formation. As a result, the ILR3-regulated iron uptake is reduced under chilling stress, which consequently attenuates the iron-overload caused damage. Under standard growth conditions, BTS finetunes free ILR3 to maintain iron homeostasis.

intron. Nevertheless, it cannot be excluded that RH24 directly regulates *ILR3* translation, given ILR3 protein was accumulated at the post-transcription level (Fig. 6f and Supplementary Fig. 6h).

Interestingly, RH24 interacts with ILR3 and is required for the nuclear body formation of ILR3, where they form phase-separated condensates (Fig. 7b and Supplementary Fig. 6k). In addition, the condensate formation is largely controlled by cold stress and BTS (Fig. 7c and Supplementary Fig. 7c). It has been reported that BTS was accumulated under cold stress[63]. Our finding links the regulatory modes of chilling sensing and iron homeostasis, which are all highly expressed in young leaves and are included in the cold-induced condensates to finetune chilling response (Supplementary Fig. 6f). In this scenario, RH24 regulates ILR3 availability and sequestering ILR3 into condensates to control iron uptake under chilling stress; BTS, on the other hand, further facilitates RH24-ILR3 condensate formation in this event to reduce iron overload. As RH24 protein levels are regulated by temperature and BTS is also responsive to temperature and iron supply, RH24 and BTS protect the plant from iron damage by maintaining iron homeostasis to cope with temperature fluctuations (Fig. 7f). In the case of *bts-2*, it is sensitive to ambient temperature, as there is a considerable amount of ILR3 accumulation (Fig. 6f and Supplementary Fig. 6h) and unlike the wild type plants, there is no BTS to finetune iron overload. It is worth of noting that the RH24-BTS-ILR3 module-mediated chilling response is largely independent of the traditional CBF pathways[10], as they were almost not affected in *bts-2* mutants under chilling stress (Supplementary Fig. 1f, g and Supplementary Data 1).

RH24 and BTS-ILR3-mediated chilling tolerance and iron homeostasis may be essential for plant survival in different geographic regions. In fact, iron toxicity often occurs in the tropic area, where the soil contains relatively higher concentrations of free $Fe^{3+}$ and $Al^{3+}$[64–66]; cold wave could be detrimental to the plants in the tropical area, assuming that RH24-like proteins were induced by cold temperature. Conversely, iron deficiency usually is observed in low-temperature areas[67]. In particular, the ecotype Col-0 used in this study is chilling tolerant and can be distributed in the geographic cold regions[68,69], where RH24-mediated iron uptake may be indispensable for the iron homeostasis of the plants in low-temperature region. Whether the RH24/BTS/ILR3-like module exists in other plants and can also finetune iron homeostasis to help plants cope with temperature fluctuation is worth of investigating in the future.

In summary, we discovered that maintaining iron homeostasis is indispensable for the plant to survive under chilling stress, and excessive application of iron could lead to low-temperature sensitivity in plants. Notably, iron accumulation-caused plant chilling sensitivity is not related to known chilling/cold-responsive CBF pathways. It will be interesting to investigate the interaction of iron homeostasis and CBF-responsive pathways. Future work will also need to focus on elucidating the comprehensive roles of RH24 and RH24-like proteins, which may be able to fully understand how plants survive in a cold environment.

## Methods

### Plant material and growth conditions

*Arabidopsis thaliana* T-DNA insertion mutants *rh24-1* (Salk_144439), *rh24-2* (Salk_045730), *rh7-2* (Salk_060686), and *ilr3-2* (Salk_004997) were obtained from the ABRC (www.Arabidopsis.org). Homozygous *bts-2* has been described previously[41]. *rh24-1/bts-2* and *rh7-2/bts-2* were generated by crossing *rh24-1* or *rh7-2* with *bts-2*. *ILR3-ox/rh24-1* was obtained by crossing *35S:ILR3-FLAG* plants with *rh24-1*. The plants were grown at 22 °C with a 10 h-light/14 h-dark cycle. For plate-growing plants, the seeds were stratified at 4 °C for 3-4 days in dark, then surface sterilized and sown on Fe-replete medium [half-strength of Murashige and Skoog (½MS) medium containing 0.05% MES, 1% sucrose, 0.4% agar, and 100 μM Fe(II)-EDTA (Ethylene diamine tetraacetic acid)

to substitute Fe sulfate], Fe-excess medium (with 300 μM Fe(II)-EDTA to substitute Fe sulfate in ½ MS media and 200 μM Fe(II)-EDTA in soil), Fe-depleted medium (with 10 μM Fe(II)-EDTA to substitute Fe sulfate) or Fe-deficiency medium (with 300 μM ferrous chelate ferrozine to substitute Fe sulfate).

### Genetic mapping and cloning of the *BTS-R* gene

Over 4000 *bts-2* seeds were treated with 0.4% EMS for 9 h, and M2 plants were screened at 16 °C. To map the *bts-r* mutation site, approximately 500 F2 seedlings were backcrossed with L*er*. Genomic DNA from these F2 plants was extracted and used for PCR-based mapping with simple sequence length polymorphism (SSLPs) markers. The DNA makers were designed based on insertions/ deletions identified from Cereon Arabidopsis polymorphism and L*er* sequence collection and were used to analyze recombination events[70]. In parallel, the F2 progeny derived from *bts-r* were backcrossed to *bts-2*. The green (*bts-r*) and the yellow (*bts-2*) plants were selected for whole-genome mapping. Genomic DNA from 20 *bts-r*-like plants and 20 *bts-2*-like plants were bulked, sequenced and assembled with their parental genomes by Illumina High-Seq and MutMap analysis[71]. The single-nucleotide polymorphisms (SNPs) for *bts-r* and *bts-2* were used for the MutMap pipeline analysis. The ΔSNP index was calculated by subtracting the *bts-2*-like bulks SNP index from the *bts-r*-like bulks SNP index. The positive Δ SNPs index values indicated the causal mutation.

### Plasmid construction and plant transformation

To generate *rh24*-complemented plants, the RH24 genomic DNA and its promoter were cloned into pGWB13 binary vector. The mutated version of RH24 (RH24^DAAD) was obtained by overlap extension PCR with primers listed in Supplementary Data 2. These constructs were transformed into *bts-r* or *rh24-1/bts-2* mutants. To generate *Cas9-RH24/bts-2* lines, three individual CRISPR/Cas9 targets in the *RH24* locus were designed through the website http://www.e-crisp.org/E-CRISP/. The sgRNA cassettes were inserted into the *pCAMBIA1300-pYAO:Cas9* binary vector[72] and was transformed into *bts-2* mutants. To generate the *RH24-FLAG* overexpression plants, the coding sequence of *RH24* was cloned into the pJL12 binary vector driven by a CaMV 35S promoter. The transformation followed the floral dipping procedure[73].

### Iron and chlorophyll concentration analysis

Leaves of four-week-old plants grown in soil were used for iron concentration measurement. The iron concentration was determined according to the method described by Xing et al.[41]. Briefly, about 2-3 g fresh leaves were dried at 80 °C for 2 days, and 0.2 g dry tissue was sampled and digested with 6 ml nitric acid and 2 ml 30% hydrogen peroxide in microwave digestion system (ETHOS UP, Milestone, Italy). The iron concentration was measured by inductively coupled plasma-mass spectrometry (ICP-MS, Hitachi, Japan). For the chlorophyll concentration assays, total chlorophylls were extracted from fresh leaves soaked in 80% acetone at 4 °C in the dark for 24 h. The chlorophyll concentration was determined by a spectrophotometer[74].

### Ion leakage, survival rate, and hypocotyl elongation analysis

The electrolyte leakage assays were performed as described previously[75]. To measure the ion leakage, plants were grown at 22 °C for 2 weeks and treated with 4 °C for indicated time, and ten uniform leaf disks were sampled and kept into 50 ml tubes containing 10 ml ddH$_2$O and incubated at 22 °C for 15 min with gentle shaking, and the conductivity was measured as S1. Then the tubes were kept into boiled water for 15 min and shaken at 22 °C for 1 h, and S2 was detected. The (S1-S0)/(S2-S0) was used to calculate ion leakage (S0: conductivity of ddH$_2$O). For chilling tolerance assays, two-week-old plants were grown at 22 °C and treated with 4 °C for 14 d. Then the plants were recovered at 22 °C for 7 d, and their survival rates (percentage of living plants) were determined. For the hypocotyl elongation assays, the seeds of

each genotype were grown on agar plates containing ½MS medium. The agar plates were covered with aluminum foil and kept vertically at 4 °C or 22 °C in growth chambers for the desired time, as described by Guan et al.[23]. Reduction in hypocotyl elongation at 4 °C compared with that of the wild type at 22 °C was used as an indicator of sensitivity to chilling stress.

### Trypan blue and DAB staining
Trypan blue staining was performed according to procedures described by Rate et al.[76] with slight modifications. Briefly, fresh leaves were boiled in equal volumes of trypan blue staining solution [0.25 mg/ml trypan blue in lactophenol (lactic acid/glycerol/liquid phenol/deionized water, 1:1:1:1)] and ethanol for 5 min, then washed in 2.5 g/ml chloral hydrate for 10 min at room temperature until complete destaining.

DAB (3,3′-diaminobenzidine) staining was performed as described by Daudi and O'Brien[77]. Leaves were collected in DAB staining solution (1 mg/ml DAB in 10 mM $Na_2HPO_4$ and 0.05% Tween 20). Then, the leaves were vacuum infiltrated for 5 min and gently shaken for 6 h at 80–100 rpm. Following the incubation, samples were transferred to the bleaching solution (ethanol/acetic acid/glycerol, 3:1:1) and boiled in boiling water for 20 min. After boiling, leaves were incubated in the fresh bleaching solution at room temperature until completely devoid of chlorophyll. The stained leaves in 10% glycerol were photographed under a stereomicroscope with a CCD camera (Olympus BX51, Tokyo, Japan).

### Molecular beacon helicase assay (MBHA)
A fluorescence-based molecular beacon helicase assay was performed as described by Belon and Frick[43]. RNA oligonucleotides were synthesized from GenScript Biotech Corporation (Nanjing, China), and were modified with a fluorophore Cyanine 5 (Cy5) on the 5′ end and Black Hole Quencher (BHQ-2) on the 3′ end. The sequences of oligonucleotides used in this study are listed in Supplementary Data 2. The duplex RNA substrates were obtained by combining labeled and unlabeled oligonucleotides at a 1:1.5 molar ratio in 20 mM Tris-HCl (pH 7.5) and 0.5 mM $MgCl_2$ by brief heating at 95 °C, followed by slow cooling to room temperature over 1 h. The unwinding reaction contained 25 mM MOPS (pH 6.5), 8 nM dsRNA substrates, 1 µM RH24 (or RH24$^{DAAD}$), 2 mM $MgCl_2$, 2 mM DTT, 0.1 mM BSA, 20 U/mL RNasin and 1 mM ATP, and the assays were carried out in 200 µl of the quartz semi-trace fluorescent cuvettes at 22 °C. Fluorescence was detected for excitation/emission at 643/667 nm using a fluorescence spectrophotometer (Hitachi F-7000, Japan) and was recorded as arbitrary units (arb. units).

### RNA duplex unwinding and ATPase activity assays
The dsRNA unwinding activity assays were determined according to procedures described by Jankowsky and Putnam[78]. The partial RNA duplex was generated by annealing an 84-nucleotide RNA with a 37-nucleotide 3′ end biotin-labeled RNA oligonucleotide as described[79]. 20 µL reaction solution contains 20 mM HEPES-KOH (pH 7.5), 2 mM $MgCl_2$, 2 mM ATP, 2 mM DTT, 50 mM KCl, 0.1 mg/mL BSA, 20 U/mL RNasin, and 0.2 pM helicase substrate. The purified recombinant proteins were incubated at 37 °C for 50 min, and then were separated on an 8% native polyacrylamide gel. The dsRNA unwinding activity of recombinant proteins was detected using Chemiluminescent Nucleic Acid Detection Module Kit (Thermo Scientific, catalog # 89880) and visualized on Tanon-5200 Chemiluminescent Imaging System (Tanon Science & Technology).

The ATPase activities of RH24 and RH24$^{DAAD}$ were determined as described by Guan et al.[23]. Briefly, the assays were performed in the reaction buffer containing 20 mM Tris-HCl (pH 7.5), 5 mM $MgCl_2$, 1 mM DTT, 50 mM KCl, 300 µM NADH, 2 mM phosphoenolpyruvate (PEP), 3 µ/mL pyruvate kinase, 3 µ/mL lactate dehydrogenase, 1 mM ATP, and

1 µM RH24 (or RH24$^{DAAD}$) with or without 50 µg/mL Arabidopsis total RNA. The decrease in the absorbance at 338 nm was detected by a multimode plate reader (Tecan Infinite F200), which reflects the ATPase activity of RH24.

### Quantitative real-time PCR (RT-qPCR) analysis
Plants were grown on ½MS medium for two weeks and treated with 4 °C for 2 days. Total RNA was extracted using Trizol Reagent (Invitrogen, catalog # 15596018). One µg of total RNA was subjected to synthesize the cDNA using the HiScript II reverse transcriptase (Vazyme Biotech Co.,Ltd, R223). CFX96™ Real-time PCR System (Bio-Rad, USA) was used to quantify the gene transcription with the ChamQ SYBR qPCR Master Mix (Vazyme Biotech Co.,Ltd, Q311). *ACTIN2* was used as a reference gene. All primers used for RT-qPCR are listed in Supplementary Data 2.

### RNA immunoprecipitation (RIP) assays
RNA immunoprecipitation (RIP) assay was performed as described by Martianov et al.[80], with slight modifications. Briefly, two-week-old transgenic *RH24-FLAG-ox* plants grown on ½MS medium were used for this experiment. Approximately 3 g of fresh leaves were harvested and crosslinked using 1% formaldehyde for 15 min. The cross-linking reaction was stopped by adding glycine to a final concentration of 125 mM for 5 min. The leaves were then ground in liquid nitrogen and extracted in RIP lysis buffer [50 mM Tris-HCl (pH 8.0), 10 mM EDTA, 1% Triton X-100, 0.1 mM PMSF, proteinase inhibitor cocktail (Roche, catalog # 04693132001) and 160 µ/mL RNaseOUT recombinant ribonuclease inhibitor (Invitrogen, catalog # 10777-019)] and sonicated 10 times for 10 sec. Samples were diluted and adjusted to 20 mM Tris-HCl (pH 8.0), 2 mM EDTA, and 150 mM NaCl. The RNA-protein complexes were immunoprecipitated by incubating with anti-FLAG M2 affinity gel (Sigma Aldrich, catalog # A2220) and protein A/G agarose beads (Thermo Scientific, catalog # 20421) at 4 °C for 6 h. The IgG was used as an IP control. The extracted input RNAs and IP RNAs were mixed with Trizol Reagent (Invitrogen, catalog # 15596018), and the precipitated mRNA was further determined by RT-qPCR. The primers used are listed in Supplementary Data 2.

### Protein interaction assays
For the immunoprecipitation (IP) assays, the CDS of *RH24* and *BTS* were cloned into pMD1-T7 and pMD1-FLAG vectors, respectively. Constructs were expressed in *N. benthamiana* leaves by *Agrobacteria*-mediated transient expression. 0.5 g infiltrated leaves were sampled for IP assays according to the method described by Xing et al.[41]. The proteins were revealed by immunoblot analysis using anti-FLAG (Sigma-Aldrich, catalog # F3165) and anti-T7 (Abcam, catalog # ab9115) antibodies.

For the split-luciferase complementation assays, the indicated nLUC and cLUC constructs were transformed into *Agrobacterium tumefaciens* strain C58C1, and then the strains were mixed at a final OD600 of 0.4 and were infiltrated into *N. benthamiana* leaves. The infiltrated leaves were rubbed with 0.5 mM luciferin 48 h later, and the signals were recorded by a Plant Imaging System with a CCD camera (NightShade LB985, Berthold technologies).

For the BiFC assays, the CDS of *RH24* and *RH24$^{DAAD}$* were cloned into the pSAT1-cYFP vector, and the CDS of *BTS* was cloned into the pSAT1-nYFP vector. These constructs were transformed into *A. tumefaciens* strain C58C1, and transiently expressed in *N. benthamiana* leaves for 48 h. The fluorescence was detected by a confocal microscope (Leica Model TCS SP8).

### Protein transient expression in Arabidopsis protoplasts
The 5′UTR-CDS-3′UTRs of *BTS* and *ILR3* were cloned and fused with a T7 epitope into the pUC19 vector at Kpn I/BstB I sites, and the CDS of *GUS*, *RH24*, and *RH24$^{DAAD}$* were cloned and fused with a FLAG epitope

into pUC19 vector. These constructs were transformed into Arabidopsis protoplasts and expressed for 16 h. The proteins were extracted using Laemmli buffer [0.0625 M Tris–HCl, 10% glycerol, 2% SDS (Sodium dodecyl sulfate), 0.0025% bromophenol blue, 5% 2-mercaptoethanol, pH 6.8] and were separated by 12% SDS-PAGE. The protein levels were detected using anti-FLAG (Sigma-Aldrich, catalog # F3165), anti-T7 (Abcam, catalog # ab9115) and anti-Actin (EASYBIO, catalog # BE0027) antibodies, respectively.

## Fluorescence recovery after photobleaching (FRAP)

FRAP were performed using a Leica SP8 laser scanning confocal microscope. The cYFP-RH24 and nYFP-ILR3 were co-expressed in *N. benthamiana* leaves by *Agrobacteria*-mediated transient expression, and then the interactions were observed under a confocal microscope (Leica SP8) at 48 hpi. At 2 h before FRAP assays, the *N. benthamiana* leaves were treated with 10 mM LatB (Abcam, catalog # ab144291) to inhibit the movement of RH24-ILR3 nuclear bodies. Then the nuclear bodies were bleached three times with a 514 nm laser at 100% laser power. The time-lapse scan modes were used for acquiring the recovery images. The FRAP data analysis were conducted as described by Boeynaems et al.[81].

## RNA-seq analysis

RNA-Seq analysis was modified from the method described by Zhao et al.[82]. Briefly, 2 g leaves were collected and used for RNA-seq library preparation. After raw data cleaning, the paired-end clean reads were aligned to the Col-0 genome TAIR10 using HISAT2 (version 2.2.1)[83]. Next, the FPKM (Fragments Per Kilobase per Million) of genes was calculated using featureCounts (version 2.0.1) and an in-house R script for further analysis[84]. The CBF-regulated genes were selected from an earlier study by Zhao et al.[13]. FPKM values in Col-0 and *bts-2* at 4 °C are provided in Supplementary Data 1. Iron-regulated genes were selected from an earlier analysis of iron deficiency-induced and repressed changes in the Arabidopsis transcriptome[42] (fold induction ≥1.5). FPKM values in Col-0 at 22 °C and 4 °C are provided in Supplementary Data 3.

## Analysis of bacterial growth

*Pseudomonas syringae* pv. *tomato* (*Pst*) DC3000 was grown on NYGA medium at 28 °C for 2 d. The leaves of four-week-old plants were syringe-infiltrated with $5 \times 10^4$ cfu/mL *Pst* DC3000 in 10 mM MgCl₂. Bacterial titers were determined by growth curve analysis at 4 days post inoculation as described by Liu et al.[85].

## Statistical analysis

All data were analyzed using one or two-way ANOVA (Analysis of variance) with GraphPad Prism software. The values are represented as means ± standard deviation (SD).

## Accession numbers

Sequence data from this article can be found in the Arabidopsis Genome Initiative or GenBank/EMBL databases under the following accession numbers: RH24 (At2g47330), BTS (At3g18290), ILR3 (AT5g54680), and RH7 (At5g62190).

## Reporting summary

Further information on research design is available in the Nature Portfolio Reporting Summary linked to this article.

## Data availability

The data supporting the findings of this work are available within the article and the supplementary information files. The genetic materials used in this study are available from the corresponding authors upon request. The RNA-seq sequencing data generated in this study have been deposited in the database of Genome Sequence Archive by China National Center for Bioinformation under accession number CRA024591. Source data are provided in this paper.

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

## Acknowledgements
We thank Prof. Hongbin Wang at Guangzhou University of Chinese Medicine for offering us the *ILR3-ox* transgenic seeds. We thank Dr. Christian Dubos at the Univ Montpellier for helping on the experiment of *ilr3-1*. We also thank Prof. Vijai Bhadauria at China Agricultural University for reading the manuscript. This work was supported by the Natural Science Foundation of China (32200246 for Y.X., 32225043 for J.L., 32270297 for N.X.) and China Postdoctoral Science Foundation (2022M713401 for Y.X.).

## Author contributions
J.L. and Y.X. conceived and designed the experiments; Y.X., Y.L., and X.G. performed the experiments; X.Z. and Q.Z. performed the RNA-seq analysis; Q.H. helped with the data analysis; Y.Q. helped with the helicase activity analysis; N.X. helped with the genetic mapping; Y.X. and J.L. wrote the article.

## Competing interests
The authors declare no competing interests.
