## [Transparent Peer Review file · Nature Communications]

An RNA helicase coordinates with iron signal regulators to alleviate chilling stress in Arabidopsis

Corresponding Author: Professor Jun Liu

Version 0:

Reviewer comments:

Reviewer #1

(Remarks to the Author)

The manuscript by Xing et al. explores the role of a DEAD-box RNA helicase, RH24, in Arabidopsis, particularly in chilling stress and iron homeostasis. The researchers employed a combination of genetic, molecular, and biochemical approaches, including suppressor screening, protein interaction assays, and various physiological and biochemical measurements, to investigate the roles of RH24 and BTS in chilling stress and iron regulation. The results indicate that (a) the BTS mutant, *bts-2*, is sensitive to chilling stress due to excessive iron accumulation, (b) through suppressor screening of *bts-2*, the DEAD-box RNA helicase RH24 was identified as a factor that suppresses the chilling sensitivity of *bts-2*, (c) RH24 binds ILR3 (a bHLH transcription factor) mRNA, (d) RH24 is required to increase ILR3 protein levels, and (e) RH24 and ILR3 form phase-separated condensates, which are further facilitated by BTS and cold treatment. The study reveals that RH24, ILR3, and BTS coordinate to regulate iron homeostasis and alleviate chilling stress in Arabidopsis. While the study seems well conducted, it is not entirely convincing. My comments are as follows: RH24 binds to ILR3 mRNA, but the cellular function of RH24 is still unknown. This manuscript lacks evidence to support the hypothesis that RH24 plays a role in ILR3 mRNA translation. Subcellular localization of RH24 seems to be in the nucleus based on the BiFC results. It is unclear how RH24 relocates to the cytosol to bind ILR3 mRNA and regulate its translation. Therefore, the subcellular localization of RH24 alone needs to be examined, particularly in plants under 22°C and 8°C conditions. Additionally, regarding subcellular localization, are ILR3 and BTS alone located in the nuclear bodies under 8°C without coexpression of RH24? According to the working model shown in Fig 7F, why do RH24-OX lines not exhibit phenotypes under chilling conditions? The results from the RH24-OX lines are not satisfactory. For instance, what is the expression level of ILR3 in RH24-OX transgenic lines? Furthermore, what are the subcellular localizations of ILR3 and BTS in RH24-OX protoplasts? These results would provide gain-of-function evidence for the working model. Other minor comments:

A. What is the expression pattern of the RH24 gene? Is it induced by chilling?

B. How does RH24 bind to ILR RNA in plants under 8°C conditions, as shown in Fig. 6C?

C. In Fig. 4F and 7F, what does mRNA refer to? If it represents RH24 and ILR3 mRNA levels, respectively, they appear overloaded (or over-cycled) for RT-PCR). Similarly, the Actin proteins used as internal controls also appear overloaded.

D. Line 364: should it be "in vitro" or "in vivo"?

Reviewer #2

(Remarks to the Author)

In the present manuscript, Yingying Xing and colleagues report on a putative mechanism linking Fe homeostasis regulation and cold stress tolerance. Mutants for BTS, which overaccumulate Fe due to a defect in Fe sensing, were more sensitive to low temperature-induced stress than WT. While exposure to 4°C significantly altered the expression of many Fe deficiency-regulated genes in WT plants, the expression of cold-responsive genes was comparable in *bts-2* and WT. By performing a mutant suppressor screen, the authors identified a mutation that largely prevented Fe overaccumulation in *bts-2* and almost fully restored to cold tolerance. The mutation was mapped to the previously uncharacterized DEAD box RNA helicase RH24. Expression of a functional RH24 in *bts-r* and *rh24-1/bts-2* double mutant plants had a WT-like phenotypes. With the recombinant protein, the authors then show that RH24 can mediate the expected ATP-dependent RNA duplex unwinding. RH24 is demonstrated to physically interact with BTS and the BTS-interacting protein ILR3 but to significantly increase only the accumulation of the ILR3 protein, likely by unwinding ILR3's mRNA. Finally, the authors show that RH24 and ILR3 colocalize in phase-separated condensates found in the nucleus.

The presented findings are novel and point to an intricate mechanism that may regulate Fe homeostasis and cold tolerance

in plants. For the most part, the experiments are well-performed. However, some characterizations remained incomplete and additional critical experimental support is still required to strengthen the conclusions. Actually, although large number of datasets are presented, there are still too many open ends that need to be connected to support the proposed BTS- and RH24-dependent chilling tolerance mechanism.

Major points:

- 1) From the study it is not clear whether BTS, RH24 and ILR3 do indeed play a specific role in cold tolerance. For instance, is the increased cold sensitivity of the *bts-2* mutant simply due to the higher Fe ROS accumulation in leaves (see Fig. 2B)? The high ROS could already predispose leaves to react more sensitively to a cold stress (e.g., increased membrane permeabilities). Actually, can the cold sensitivity of *bts-2* plants be partially rescued by ROS-scavengers and would any high ROS-producing mutant naturally be more prone to chilling injury?
- 2) How can the authors be certain that the increased sensitivity of *bts-2* plants to low temperatures is associated with the Fe overaccumulation phenotype and not with another process regulated by the putative E3 ligase function of BTS? Actually, to prove that the increased cold sensitivity of *bts-2* plants is due to Fe overaccumulation, the latter should be prevented in the *bts-2* background, for instance by generating a *bts-2 irt1* double mutant. According to Fig. 8D, at 10 μ M external Fe, *bts-2* plants were apparently smaller than WT at 8°C but not at 22°C. But were the shoot Fe concentrations comparable at this low Fe supply? Was the small shoot size indeed a phenotype of cold stress?
- 3) Furthermore, the authors show that RH24 protein accumulation in RH24-overexpressing plants was increased at low temperature (Fig. S4B). However, is RH24 (transcripts/protein levels) indeed temperature-responsive in the wild-type background?
- 4) Although BTS, ILR3 and RH24 are shown to interact when transiently expressed in tobacco leaves or Arabidopsis protoplasts, no sufficient evidence is provided that the proteins are present in the same tissues, especially under the low temperature condition. At least the tissue-specific expression pattern of RH24, preferentially under different temperatures, must be determined and compared to BTS and ILR3.
- 5) Experimental evidence is lacking to demonstrate that BTS facilitates RH24-BTS condensate formation under low temperature. Can this effect be presented beyond the transient assays in tobacco leaves? Furthermore, according to Fig. 1A, BTS expression is down-regulated by 4°C in WT. Thus, how can it positively regulate condensate formation at low temperatures as proposed in the model (Fig. 7F)?
- 6) The evidence for the involvement of RH24 in the regulation of Fe deficiency responses and chilling tolerance is weak, as it mainly originates from disrupting RH24 in the *bts-2* background. To strengthen this conclusion, shoot Fe concentration and quantitative assessment of chilling tolerance must be systematically performed with independent *rh24* single mutants and RH24 overexpressors.
- 7) Furthermore, regarding the link between Fe and chilling tolerance, it remains to be demonstrated whether disruption of RH24 results in less Fe accumulation at high Fe supply, when the mutant apparently tolerates better the 4°C treatment and has less ion leakage than WT (Fig 5B-C). If the quantitative differences in shoot Fe are as small as in Fig. 5A, then it is hard to believe that Fe accumulation can explain the increased cold tolerance of the *rh24-1* mutant. In my view, besides biomass and shoot Fe quantification, it is important that experiments as shown in Fig. 5B and C are carried out with two independent *rh24* mutants.
- 8) In order to strengthen the proposed link between Fe accumulation and chilling tolerance, additional experimental evidence is necessary. For instance, Fig. 1C shows that Fe concentration of *bts-2* plants can almost recover back to WT when grown at 26°C. Is this related to altered biomass (change in dilution factor) or altered regulation of changes in the expression of deregulated genes in *bts-2*? Furthermore, without biomass quantification and shoot Fe concentration data it is not possible to conclude that “high content of iron in the plants indeed impaired their chilling tolerance, especially *bts-2* plants”, as done in lines 194-195.

Specific points:

- 9) In general, the manuscript is well-written. However, additional language is editing to eliminate typos and, especially, to improve grammar.
- 10) Considering the importance of IRT1, FRO2, AHA2 and the coumarin-related genes F6'H1, S8H and CYP82C4 in root Fe uptake, the expression of these genes should be also presented.
- 11) Please describe more clearly if whole plants or only roots were sampled for transcriptome reported in Fig. 1A.
- 12) In plant physiology, the unit of nutrient/chlorophyll etc per unit of biomass is “concentration”, whereas “content” is reserved for unit per organ. Please correct in all figures accordingly. Furthermore, indicate if Fe concentration is on fresh or dry weight basis and whether whole shoots or selected leaves (and if so, which ones) were sampled for elemental analysis.
- 13) Fig. 1E: from the plot, it is hard to understand to what comparisons the delta values refer to.
- 14) What was the statistical test employed in Fig. 6E?

Reviewer #3

(Remarks to the Author)

The article by Zhao et al. describes the role of the helicase RH24 in the regulation of ILR3 under chilling stress and links this function to the control of Fe homeostasis. In my opinion, the results provided are relevant to the field. In general terms, the authors present multiple experiments to test their hypotheses, which makes the results seem solid. Especially notable is the phenotypic characterization of the different mutants regarding chilling and Fe sensitivity. Nevertheless, I strongly think that the authors should provide more experiments if they want to propose that RH24 controls ILR3 translation. These and other concerns are described here:

- I have some concerns about the experiments carried out with the *bts* mutants. In most cases, the Col-0 and the *bts-2* mutant were grown at 22°C and then subjected to chilling. However, the *bts* mutant seems to have a developmental phenotype at

22°C (Figure 2A). If there are significant differences in development between Col-0 and the *bts-2* mutant at the time of applying the challenge, how can the authors be sure that the differences are due to chilling sensitivity and not because the *bts-2* plants are already smaller when challenged at 4°C? It is established that the developmental defect associated with *bts-2* is not visible at 26°C. To avoid this problem, they could grow the plants at 26°C and challenge them at 4°C.

- Since the authors have RNAseq data regarding the *bts-2* mutant at 22°C and 4°C, it would be helpful to generate a figure similar to the one displayed in Figure 1A to highlight the differences in the expression of Fe-responsive genes in the *bts-2* mutant at both temperatures.
- In Figures 6A-B, the control of the RH24-cYFP is missing. In addition, could authors provide more information about the protein (EV) they use in the negative control? Is it localized in the nucleus?
- In Figures 6D-F, the authors observe that in the presence of RH24, ILR3 accumulates, and that this accumulation is reduced in the *rh24* mutant. Since RH24 seems to bind to ILR3 RNA, they hypothesize that this could be due to an enhancement of translation. However, since RH24 and ILR3 proteins interact, it is also possible that in the presence of RH24 and/or cold, the ILR3 protein could be more stable and be degraded by the proteasome in the absence of RH24. I suggest conducting the experiments in the presence of proteasome inhibitors. Additionally, they should directly probe if ILR3 is regulated at the translational level by RH24. There are different procedures, such as polysome gradients, sucrose cushions, and ribo-Seq, that could be done in the wild type and *rh24* mutant at 22°C and 4°C. A genome-wide translation study on the *rh24* mutants could also provide additional targets of RH24 translational regulation. These experiments could also be done to assess how RH24 accumulates under chilling conditions, since RH24 accumulates at 4°C, but its transcription doesn't seem to change, RH24 could also be a good candidate to be regulated at the translational level.
- Have the authors included the 5'UTR and/or the 3'UTRs of BTS and ILR3 in the 35S: BTS-T7 and p35S: ILR3-T7 constructs in Figures 6D-F? It has to be considered that in many cases the translational regulatory elements are present in the UTRs and that these elements could be essential to observe the translation-dependent accumulation of these proteins. Taking into account that RH24 is an RNA helicase and, therefore, could have a role in translation regulation, without these sequences, for example, they cannot fully discard in Figure 6B that RH24 is involved in regulating BTS accumulation. Additionally, it might be possible that the addition of these sequences could provide different results regarding the BTS and ILR3 accumulation.
- In Figures 6D, 6F, S4, the authors provide a quantification of a single experiment.. Is it possible to quantify and show the statistics of the whole set of three independent experiments? Additionally, I have not seen what mRNA stands for in these figures; I suppose it is a semiquantitative PCR of the ILR3, BTS genes, etc. Authors should provide the statistics of the whole set of experiments and the analysis of expression by qRT-PCRs, if possible.
- Figure 7. To probe that RH24 controls the formation of ILR3 condensates, I suggest analyzing the nuclear condensates containing ILR3 in the *rh24* mutant and RH24 OE plants? In addition, I miss the information about the role of RH24 in Fe sensitivity in the model. Please include it.
- Have author tested the expression of the ILR3 responsive genes in *rh24* and RH24 OE lines? Since ILR3 is a TF it would be interesting to analyze the expression of the ILR3 responsive genes to further support the hypothesis.
- In Figure S1 E-F, considering the legend, it seems that E and F are different ways to present the level of expression of the CBF-responsive genes in the *bts-2* mutant at control and at 4°C. However, it seems to me that in F, most of the significant changes occur at 0 h, while in E, the most significant changes occur at 3 h and 24 h.

Minor points:

- The authors do not follow a logical order to present the data in the Figures. For example, in Figure S1, G panel is between C and E panels, and this also happens in other figures. I think it would facilitate the reading if the panels in the figures are logically presented.
- In Figure 2B, the legend does not clearly indicate whether these specific experiments were carried out under chilling or control conditions.
- Some paragraphs are difficult to follow. For example, the paragraph in lines 159-165.
- When referring to 4°C conditions, I would rather use chilling sensitivity than temperature sensitivity.
- In line 250, there should be a mistake with the panels in the Figure S3, since figure S3A does not show the chilling and iron deficiency phenotypes of *bts-2*.
- For clarity, in the description of the *bts-r* lines expressing RH24 (DAAD), I would rather refer to "phenocopy the *bts-2* mutant."

Reviewer #4

(Remarks to the Author)

In their work, Xing et al. provide compelling evidence demonstrating the DEAD-box RNA helicase RH24 acts as a molecular connection between iron homeostasis and chilling stress in *Arabidopsis thaliana*. Their experiments using the BRUTUS (BTS) null mutant *bts-2*, which displays increased iron content and therefore is resistant to iron deficiency, revealed that this mutant was sensitive to low temperatures (8°C and 4°C). Interestingly, the authors performed a genetic screening, which revealed that mutation of a DEAD-box RNA helicase RH24, was sufficient to suppress the chilling stress phenotype observed in *bts-2*, i.e. stunted growth and lower chlorophyll content. Importantly, using an impressive set of loss of RH24 and RH24/BST function lines the authors robustly describe how RH24 regulates iron accumulation and cold stress tolerance. They also found that RH24 co-localizes with BTS and BTS-interacting proteins IL3 in the nucleus. Remarkably, they revealed that RH24 affects IL3 protein levels by controlling the unwinding of ILR3 RNA. The manuscript is relatively well-written, and the experimental evidence is robust, however, I have a few suggestions.

#Figure 1A and Figure S1E. The authors show the expression of the iron metabolic pathway DEGS. In line with the results, they observed high expression of BTS in Col-0 plants at 26 and 4 °C, and a different pattern of CBF genes between Col-0 and *bts-2* mutant. In base on these results, one would expect to observe a differential expression for ILR3 and the helicase

gene family under the same conditions. Have the authors looked at these specific genes?

#Figure 1D. Based on this result it is rather clear that *bts-2* displays an increased sensitivity to cold (above 0 °C). Due to the stunted growth of this mutant at 8 °C, it is hard to see an effect of the iron concentration on the mutant growth. I think showing the same experiment with an additional cold acclimation period would help to understand these results.

#2: Line 189-190: Throughout the manuscript, the authors used the term "enhanced temperature sensitivity." I recommend replacing "enhanced" with "increased."

#Figure 4F and Figure 6F: In general, the quality of these images is low. I strongly recommend showing the protein concentration levels of at least three independent replicates showing the error bars. Additionally, to support the authors' conclusion that RH24 accumulates under chilling conditions in Figure 4F, I suggest including an additional western blot using cycloheximide to support the authors' conclusion that RH24 accumulates under chilling conditions

#Figure 7F. I found the scheme hard to understand and interpret. Moreover, I was a bit surprised that the final model does not integrate together cold and iron homeostasis

Version 1:

Reviewer comments:

Reviewer #2

(Remarks to the Author)

The authors have addressed all points raised in my initial review, providing new experimental data and clarifications in the text. The new data and the amendments in the text have improved the manuscript. However, some points were not yet addressed satisfactorily and need further clarifications and even more experimental evidence, as outlined below.

Major points:

1) To reply to my comment of whether the increased cold sensitivity of the *bts-2* mutant simply due to the higher Fe ROS accumulation in leaves, the authors performed one experiment in which they show that supply of ascorbic acid did not rescue the cold sensitivity of *bts-2*. Furthermore, they refer to other studies to support their conclusion that the chilling sensitivity of *bts-2* was not due to the high ROS accumulation in this mutant. This is not completely satisfactory: 1) ascorbic acid can also act as a pro-oxidant, as it can reduce Fe(III), thus evidence based solely on this compound is difficult to interpret; and 2) the conditions used in the mentioned studies with H₂O₂ supply, are not necessarily reflecting the ROS accumulation in plant tissues induced by the Fe overload in *bts-2*. Actually, there is an alternative way to directly address this critical question, which the authors did not explore. Most phenotypes of *bts-2*, including the strong Fe overaccumulation, are almost fully reverted by growing plants at 26°C (Figs. 1C and S1). Is ROS accumulation decreased at this elevated temperature? If yes, would *bts-2* plants pre-cultured at 26°C still show increased chilling sensitivity when then transferred to lower temperatures?

2) Following on that, can 26°C largely rescue the phenotypes of *bts-2* because it strongly suppresses RH24 protein accumulation at this temperature? This possibility should be at least discussed in the manuscript.

3) The experiment suggested above would also provide additional evidence to support that the increased sensitivity of *bts-2* plants to low temperatures is associated with the Fe overaccumulation phenotype and not with another process regulated by the putative E3 ligase function of BTS.

4) The fact that RH24, ILR3 and BTS are highly expressed in young leaves is very important for the whole model. However, I could not find this dataset in the manuscript. These results must be described and discussed in the manuscript.

Minor points:

5) Following on points 1 and 2, does the reversion of several phenotypes of *bts-2* at 26°C suggest that the mutant is not more sensitive to chilling but actually even to normal ambient temperature? This point should be discussed.

6) Some statements do not reflect what is shown by the data. For instance, lines 177-179, from the heatmap of Fig. 1A, only FER1 was "strongly induced" in *bts-2* compared to WT at 4°C and not FERs in general, as the sentence suggests. Please amend throughout the manuscript.

Reviewer #3

(Remarks to the Author)

I consider that authors have answered successfully to my main concerns. However, I feel that they can include in the supplementary data the figure 7 from the answer to the reviewers. I do not see how this could make more complicated the article.

Regarding the role of RH24, I feel that it is not completely revealed. However, I also consider that authors have obtained many and important pieces of information about the role of BTS, RH24 and ILR3 in cold stress and iron homeostasis and of the link between the three proteins and the two process. In my opinion, all this information makes this manuscript worth to be published in this journal.

Line 491, it is established "Instead, RH24 binds with and likely splices ILR3 mRNA, which consequently regulates ILR3 protein levels". If I am not wrong, the data regarding the role of RH24 in splicing are only included in the section: response to the reviewers, but this information is not available in the Figures to the readers. Without giving this information to the readers, I think that the sentence in line 491 could not be stated. I propose the authors to include the figure and discuss how the splicing might modulate the accumulation of the protein. I also propose the authors to include a sentence to clarify that the role of RH24 in ILR3 regulation of translation cannot be completely excluded.

Minor suggestions:

Line 79, "Except for the transcription factors, post-transcriptional regulation is also involved in chilling tolerance", I suppose that authors want to say "As long as transcription, post-transcriptional regulation is also involved in chilling tolerance.."
Line 163 "....DEGS of Col-0 plants showed remarkably reduced expression for the major iron-responsive genes, including IMAs.....". This sentence will be clearer to me if instead of major iron-responsive genes they include the categories that appear in the figure such as "low iron inducible genes, iron deficiency-induced transcription factors, etc".

What do mean rh24#5 sgRNA and rh24#2 sgRNA in Figure 3C?

Figure 4. Authors make a lot of effort to demonstrate that RH24 is an ATP dependent RNA helicase. Could they include in the discussion how this activity could modulate the splicing of ILR3?

Line 296 "... we conclude that the dsRNA unwinding activity of RH24 is indispensable for suppressing chilling sensitivity to bts-2". I would soften the sentence to "we conclude that, most probably, the dsRNA unwinding activity of RH24 is indispensable for suppressing chilling sensitivity to bts-2".

Line 340. I think that there is a mistake and they do not refer in this paragraph to Figure 5E.

I think that authors should rephrase those sentences related to the effect of MG132: Line 377, line 390 and 493, they are quite confusing. For example, in the case of phrase, line 390, "the proteasome inhibitor MG132 did not increase the accumulation of ILR3 in the ILR3-ox/rh24-1 plants, suggesting that RH24 facilitates ILR3 accumulation at transcription or translation level under chilling conditions", I would propose "the proteasome inhibitor MG132 did not increase the accumulation of ILR3 in the ILR3-ox/rh24-1 plants, suggesting that the reduced accumulation of ILR3 accumulation in the rh24-1 is not due to ILR3 protein degradation".

Line 491 "...., and bts-2 chilling sensitivity is suppressed by RH24 (Figures 3 and S6)". Do you mean: "...., and bts-2 chilling sensitivity is suppressed in the absence of RH24 (Figures 3 and S6)"?

I observed some confusing sentences in the materials and methods. For example line 690...." These constructs were mixed and transformed into *N. benthamiana* leaves by *Agrobacterium tumefaciens*,". I suppose that they want to say "These constructs were used to transform *Agrobacterium tumefaciens*,..., and the transformed strains were mixed to agroinfiltrate *N. benthamiana* leaves. This could be also read in line 693. If so, please, could go through materials and methods to describe properly the experiments.

Reviewer #4

(Remarks to the Author)

Version 2:

Reviewer comments:

Reviewer #2

(Remarks to the Author)

The authors have satisfactorily addressed all concerns. There are only a few minor amendments required as follows:

L159-160: Please, rephrase to: „Because BTS is a key regulator of Fe deficiency responses, ...

L416: Correct abbreviation to FRAP

In several plots, the dots representing the original datapoints are too small. Actually, the size of the dots is not consistent, sometimes not even within the same plot (e.g., Fig. 2F). I recommend the authors to choose a dot size that allows to clearly see the datapoints (e.g., plot of Fig. 6F) and use that consistently in all plots.

Other than that, I want to congratulate the authors for this impressive and well-executed study reporting several novel and relevant messages.

Reviewer #3

(Remarks to the Author)

I sincerely appreciate the authors' efforts in revising the manuscript. For instance, they clarified the role of RH24 in IRL3 regulation and improved the description of the discussion sections. This revision has significantly enhanced the overall quality and robustness of the manuscript. The manuscript is now suitable for publication, as it provides important insights and contributes meaningfully to the field.

Minor comments:

Comment 1: I strongly recommend that the authors include the number of plants used per replicate in their experimental design.

Comment 2: In Figure 2 C, it is stated that "...and the iron deficient condition was made by applying 300 μ M Ferrozine in $\frac{1}{2}$ MS medium to chelate iron (right panel) ..." I believe that the authors intended to refer to the lower panel instead of the right panel.

Reviewer #4

(Remarks to the Author)

Dear Editor and Reviewers:

Thanks for the constructive comments and insightful suggestions. We have tried our best to address all the concerns raised by reviewers.

Reviewer comments:

Reviewer #1:

Comment 1: RH24 binds to ILR3 mRNA, but the cellular function of RH24 is still unknown. This manuscript lacks evidence to support the hypothesis that RH24 plays a role in ILR3 mRNA translation. Subcellular localization of RH24 seems to be in the nucleus based on the BiFC results. It is unclear how RH24 relocates to the cytosol to bind ILR3 mRNA and regulate its translation. Therefore, the subcellular localization of RH24 alone needs to be examined, particularly in plants under 22°C and 8°C conditions. Additionally, regarding subcellular localization, are ILR3 and BTS alone located in the nuclear bodies under 8°C without co-expression of RH24?

Response: Thanks for this comment. We investigate the subcellular localization of RH24, BTS and ILR3 in Arabidopsis cells under 22°C or 8°C condition, respectively. We discovered that the RH24 was only located in the nucleus under 22°C or 8°C condition (below Figure 1A). Because RH24 locates in the nucleus, theoretically it is unable to regulate the translation of ILR3 in cytosol. However, we discovered that RH24 splices ILR3 mRNA, which likely occurs in nucleus and facilitates the translation indirectly (see below Figure 8). Because it is very difficult to build a link between splicing and translation, we remove the word “translation”. However, BTS alone was only located in the nucleus but not the nuclear bodies in Col-0, *rh24*, and *RH24-ox-3* plants under 8°C (below Figure 1B), and ILR3 alone was able to locate in the nuclear bodies in the presence of RH24 under 8°C, but not in *rh24* mutant where no nuclear bodies could be observed (below Figure 1C), supporting our conclusion that ILR3 nuclear body formation requires RH24. Based on the result that BTS facilitated the RH24-ILR3 condensate formation (Figure 7C and S7C), we conclude that BTS and RH24 facilitate the ILR3 nuclear body formation.

Fig 1. The subcellular localization of RH24, BTS and ILR3 alone under 22°C or 8°C condition in Col-0, *rh24* mutant and RH24-ox plants.

(A) RH24 was only located in the nucleus under 22°C (control) or 8°C (cold) condition in Col-0.

(B) BTS alone was only located in the nucleus, but not the nuclear bodies. BTS was only located in the nucleus in *rh24* mutant or RH24-ox transgenic plants under 8°C (cold) condition.

(C) ILR3 was located in the nuclear bodies in Col-0 and RH24-ox transgenic plants under 8°C (cold), but not in *rh24* mutants.

The CDS of RH24, BTS and ILR3 were cloned into pSAT6-EYFP-N1 vector. RH24-EYFP, BTS-EYFP and ILR3-EYFP were expressed in Arabidopsis protoplast at 22°C for 12h and then transferred to 22°C or 8°C for another 1h, respectively. Nucleus was stained with DAPI. Chlorophyll was shown with autofluorescent light. BF: bright field. Bar = 3µm.

Comment 2: According to the working model shown in Fig 7F, why do RH24-OX lines not exhibit phenotypes under chilling conditions? The results from the RH24-OX lines are not satisfactory. For instance, what is the expression level of ILR3 in RH24-OX transgenic lines? Furthermore, what are the subcellular localizations of ILR3 and BTS in RH24-OX protoplasts? These results would provide gain-of-function evidence for the working model.

Response: We understand this reviewer’s concerns. Indeed, RH24-ox lines displayed a weak phenotype compared to Col-0 under chilling conditions (Figure S5D and S5E). We reason that RH24-mediated ILR3 accumulation is also controlled by BTS and the condensation of ILR3, which leads to attenuated response to chilling. A sensitive method to examine the hypocotyl elongation under chilling conditions was often used to analyze the weak response (Guan et al., 2013, Plant Cell; Liu et al., 2021, New Phytologist). We found that the hypocotyl elongation of RH24-ox lines is more sensitive to chilling stress, where the hypocotyl elongation was significantly suppressed in Fe-oversupplied conditions under 4 °C (see the revised Figure 5A). Meanwhile, there was no significant difference in the transcription levels of *ILR3* in Col-0, *rh24* and RH24-OX transgenic plants, but the protein levels were elevated in RH24-OX transgenic lines (see below Figure 2). It suggested that RH24 influences ILR3 at post-transcription level. As the above Figure 1 shows, we discovered that ILR3 was primarily located in nuclear bodies in the RH24-OX protoplasts, especially under 8°C (cold) condition, while BTS only locates in nucleus (see above Figure 1). It is known that ILR3 only locates in nucleus in *bts* mutant (Selote et al., 2015, Plant Physiology, Figure 5C), and we observed similar location (data known shown). Therefore, we concluded that ILR3 condensate formation requires both BTS and RH24.

Fig 2. The expression levels of ILR3 in Col-0, *rh24* and RH24-OX transgenic plants under 22°C or 4°C condition.

(A) The transcription levels of *ILR3* in Col-0, *rh24* and RH24-OX transgenic plants. Four-week-old Col-0, *rh24-1*, *rh24-2*, RH24-OX-3 and RH24-OX-10 plants were grown at 22 °C and then subjected to 4 °C treatment for 24 h. Significant differences were analyzed by a two-factor ANOVA with Tukey's HSD test ($p \leq 0.05$, Data are means \pm SD, n=3 biological replicates).

(B) The protein levels of ILR3 in Col-0, *rh24* and RH24-OX transgenic plants. ILR3-YFP was expressed in Arabidopsis protoplasts at 22°C for 12h and then transferred to 22°C or 4°C for another 6h, respectively.

Minor comments 3: What is the expression pattern of the RH24 gene? Is it induced by chilling?

Response: We thank this reviewer's suggestion. We found that the RH24 transcripts were detected in all selected tissues, with relatively high expression levels in young leaves and rosette leaves (see below Figure 3A). In addition, we discovered that *RH24* mRNA abundance was slightly increased at 12h and 24h under 4°C (see below Figure 3B).

Fig 3. Expression patterns of RH24.

(A) Expression levels of RH24 in various tissues. Total RNA was isolated from roots, stems, young leaves, rosette leaves, cauline leaves and flowers.

(B) RT-qPCR analysis of *RH24* expression under cold stress. Four-week-old plants were grown at 22°C and then were subjected to 4°C treatment for 3, 6, 12, and 24h.

Minor comments 4: How does RH24 bind to ILR RNA in plants under 8°C conditions, as shown in Fig. 6C? (Fig. 6E)

Response: It is very difficult to answer this question without protein crystal structure. Although many RH24 homologous helicase structures have been resolved, the mechanism for how these helicases bind RNAs is still elusive. In our case, we found that RH24 somehow can splice *ILR3* intron (see below Figure 8). Theoretically, it binds then splices the intron of pre-mRNA of *ILR3*. In addition, using the mature mRNA as template, we can immunoprecipitate the mRNA, suggesting that RH24 binds mature mRNA (Figure 6E).

Minor comments 5: In Fig. 4F and 7F, what does mRNA refer to? If it represents RH24

and ILR3 mRNA levels, respectively, they appear overloaded (or over-cycled for RT-PCR). Similarly, the Actin proteins used as internal controls also appear overloaded.

Response: We agree with this comment. In Fig. 4F and 7F (I assume 6F), mRNA refers to the *RH24* (Fig. 4F) and *ILR3* (Fig. 6F) mRNA levels, respectively. In the revision, to exclude the issue of overload, we performed long exposure and short exposure experiments and employed RT-qPCR to examine the mRNA levels, which are comparable (see the revised Figure 4F and 6F).

Minor comments 6: Line 364: should it be "in vitro" or "in vivo"?

Response: Sorry for the mistake. It is "in vivo" and it's been corrected.

Reviewer #2:

Major points:

Comment 1: From the study it is not clear whether BTS, RH24 and ILR3 do indeed play a specific role in cold tolerance. For instance, is the increased cold sensitivity of the *bts-2* mutant simply due to the higher Fe ROS accumulation in leaves (see Fig. 2B)? The high ROS could already predispose leaves to react more sensitively to a cold stress (e.g., increased membrane permeabilities). Actually, can the cold sensitivity of *bts-2* plants be partially rescued by ROS-scavengers and would any high ROS-producing mutant naturally be more prone to chilling injury?

Response: Thanks for this insightful comment. We checked the cold sensitivity of *bts-2* mutant in the presence of ROS-scavengers (ascorbic acid, AsA). AsA, as one of the universal antioxidants, plays a key role in scavenging ROS. Exogenous 0.5mM ASA could induce the chilling tolerance in tomato and spinach plants (Celi et al., 2023, Plant Physiology and Biochemistry; Wang et al., 2024, International Journal of Molecular Sciences). However, in our case, exogenous 0.5mM or 1mM ASA could not rescue the sensitivity of *bts-2* (see below Figure 4).

Regarding the chilling response of ROS-producing mutants, Liu et al. reported that the *rboh-D* and *rboh-F* mutants, which were defective in generating ROS, were more sensitive to cold stress (Liu et al., 2022, Developmental Cell). By contrast, plants treated with exogenous 100 and 200 μ M H₂O₂ showed much higher survival rate than untreated controls under cold acclimation conditions in mung bean, manila grass and tomato (Hung et al., 2007, Journal of American Society for Horticultural Science; Wang et al., 2010, Plant Growth Regulation). Therefore, we concluded that the chilling sensitivity of *bts-2* was not due to the higher ROS accumulation in leaves in our case.

Fig 4. ROS-scavengers (ascorbic acid, AsA) cannot rescue the chilling sensitivity of *bts-2*.

Col-0 and *bts-2* plants were grown on $\frac{1}{2}$ MS medium at 22°C for 7 days and then transferred to $\frac{1}{2}$ MS medium

supplemented with 0, 0.5, and 1mM AsA for 4 weeks at 8°C.

Comment 2: How can the authors be certain that the increased sensitivity of *bts-2* plants to low temperatures is associated with the Fe overaccumulation phenotype and not with another process regulated by the putative E3 ligase function of BTS? Actually, to prove that the increased cold sensitivity of *bts-2* plants is due to Fe overaccumulation, the latter should be prevented in the *bts-2* background, for instance by generating a *bts-2 irt1* double mutant. According to Fig. 1D, at 10 μ M external Fe, *bts-2* plants were apparently smaller than WT at 8°C but not at 22°C. But were the shoot Fe concentrations comparable at this low Fe supply? Was the small shoot size indeed a phenotype of cold stress?

Response: We recognize this reviewer's concern. In fact, the BTS E3 ligase controls ILR3 protein levels and the iron concentration in cells. Certainly, the E3 ligase activity is highly related to the chilling response. Although we did not generate *bts-2/irt1*, to prove this hypothesis as suggested, we communicated with the authors of Li et al. (2019, Journal of Plant Biology), who generated the *bts-2/ilr3-2* double mutant. The double mutant theoretically acts similar as *bts-2/irt1*. In fact, they found that the double mutant displayed WT-like phenotype under chilling conditions or 22°C (Li et al., 2019, Journal of Plant Biology, Fig2A), suggesting that the *bts-2* chilling sensitivity is due to overload of iron.

Under 8°C condition, our Perls staining results repeatedly showed that *bts-2* contained higher Fe concentrations than Col-0 at 10 μ M iron supply (data not shown). Regarding the shoot size, *bts-2* showed similar size as Col-0 at 22°C and 10 μ M iron supply, suggesting that the small shoot size of *bts-2* is indeed due to iron overaccumulation under chilling condition. Because these data will make the result being complicated, we choose not having them in the manuscript.

Comment 3: Furthermore, the authors show that RH24 protein accumulation in RH24-overexpressing plants was increased at low temperature (Fig. S4B). However, is RH24 (transcripts/protein levels) indeed temperature-responsive in the wild-type background?

Response: We discovered that the expression level of *RH24* was slightly induced at 4°C compared to 22°C or 26°C in Col-0 (see below Figure 5B), indicating that RH24 was slightly induced by chilling stress. It would be nice to examine the RH24 protein levels in WT. We failed to obtain reliable RH24 antibody. Nevertheless, the conclusion for the low temperature stabilizing RH24 is not compromised (Figure 4F), and the merit of our study should not be essentially affected.

Comment 4: Although BTS, ILR3 and RH24 are shown to interact when transiently expressed in tobacco leaves or Arabidopsis protoplasts, no sufficient evidence is provided that the proteins are present in the same tissues, especially under the low temperature condition. At least the tissue-specific expression pattern of RH24, preferentially under different temperatures, must be determined and compared to BTS and ILR3.

Response: Thanks for this comment. We performed the RT-qPCR assays for the tissue-

specific expression pattern and cold-induced expression pattern of RH24, BTS, and ILR3 (below Figure 5). We found that the expression of ILR3, RH24, and BTS were slightly highly expressed in young leaves (see below Figure 5A). However, their expression was slightly affected at 4°C (see below Figure 5B).

Fig 5. Expression patterns of RH24, ILR3, and BTS.

(A) Expression levels of RH24, ILR3, and BTS in various tissues. Total RNA was isolated from roots, stems, young leaves, rosette leaves, cauline leaves and flowers.

(B) Expression levels of RH24, ILR3, and BTS at 4 °C, 22 °C, and 26°C. Four-week-old plants were grown at 22°C and then subjected to 4 °C, 22 °C, and 26°C for 24h.

The actin gene was used as an internal control. Significant differences were analyzed by one-way ANOVA with Tukey's HSD test ($p \leq 0.05$, Data are means \pm SD, n=3 biological replicates).

Comment 5: Experimental evidence is lacking to demonstrate that BTS facilitates RH24-BTS condensate formation under low temperature. Can this effect be presented beyond the transient assays in tobacco leaves? Furthermore, according to Fig. 1A, BTS expression is down-regulated by 4°C in WT. Thus, how can it positively regulate condensate formation at low temperatures as proposed in the model (Fig. 7F)?

Response: As this reviewer suggested, we also performed this experiment in Arabidopsis protoplast (see the revised Figure S7C), which is consistent with the result from the transient assays in *N. benthamiana*. For BTS, the RNAseq and qRT-PCR results do show a slightly decreased expression at mRNA level (see above Figure 5B). However, according to Selote et al. report, BTS protein was accumulated at 4°C (2018, Plant Cell Environ). Therefore, we consider that BTS was regulated at protein level

under chilling conditions.

Comment 6: The evidence for the involvement of RH24 in the regulation of Fe deficiency responses and chilling tolerance is weak, as it mainly originates from disrupting RH24 in the *bts-2* background. To strengthen this conclusion, shoot Fe concentration and quantitative assessment of chilling tolerance must be systematically performed with independent *rh24* single mutants and RH24 overexpressors.

Response: To reinforce the result of RH24 in chilling response, we performed a sensitive experiment to demonstrate the chilling sensitivity of *rh24-1*, *rh24-2*, and two RH24-ox lines by the hypocotyl elongation assay, which can nicely show the plants are sensitive to cold treatment (Liu et al., 2021, New Phytologist) (see the revised Figure 5A-B). Moreover, we measured Fe accumulation in leaves of *rh24-1*, *rh24-2*, and two RH24-ox lines by Perls Fe staining (see revised Figure S5C), showing that *rh24-1* and *rh24-2* mutant leaves accumulated less Fe than Col-0 and RH24-ox lines at 4°C. Thus, we confirm that RH24 indeed regulates plant chilling and iron deficiency response.

Comment 7: Furthermore, regarding the link between Fe and chilling tolerance, it remains to be demonstrated whether disruption of RH24 results in less Fe accumulation at high Fe supply, when the mutant apparently tolerates better the 4°C treatment and has less ion leakage than WT (Fig 5B-C). If the quantitative differences in shoot Fe are as small as in Fig. 5A, then it is hard to believe that Fe accumulation can explain the increased cold tolerance of the *rh24-1* mutant. In my view, besides biomass and shoot Fe quantification, it is important that experiments as shown in Fig. 5B and C are carried out with two independent *rh24* mutants.

Response: By Perls Fe staining, we observed slightly higher difference between *rh24* mutant and WT at high iron supply than those at normal iron supply (data not shown). Because our model is not based on high iron supply, and including these data may confuse readers, we therefore do not include these data in the revision. Regarding the weak chilling tolerance of these mutants, it in fact supports Fe accumulation and chilling response as these mutants accumulated slightly less iron. As such, one cannot expect large difference on chilling tolerance. As suggested, we now included two independent mutant lines in revised Figure 5C-D.

Comment 8: In order to strengthen the proposed link between Fe accumulation and chilling tolerance, additional experimental evidence is necessary. For instance, Fig. 1C shows that Fe concentration of *bts-2* plants can almost recover back to WT when grown at 26°C. Is this related to altered biomass (change in dilution factor) or altered regulation of changes in the expression of deregulated genes in *bts-2*? Furthermore, without biomass quantification and shoot Fe concentration data it is not possible to conclude that “high content of iron in the plants indeed impaired their chilling tolerance, especially *bts-2* plants”, as done in lines 194-195.

Response: We understand this reviewer’s concerns. In fact, at 26°C, we examined several key iron regulators, where we did not observe significant differences in *bts-2* compared to that at 22°C (see below Figure 6). These data strongly suggest the iron

accumulation in *bts-2* was largely regulated at post-transcription by condensate formation. Related to the biomass and dilution issue at 26°C, the Fe content that we showed was calculated by the unit of iron per unit of dry weight leaves, which can indicate the true Fe status in plant tissues, and *bts-2* mutant accumulates more iron. As a negative control, we found *srf1-4* that is chilling sensitive mutant did not accumulate much higher iron and did not exhibit cell death compared to Col-0, suggesting that dilution issue may not apply to *bts-2* in our case.

Fig 6. The expression levels of *ILR3*, *bHLH115*, and *PYE* under 22°C or 26°C.

Four-week-old plants were grown at 22°C and then subjected to 22°C or 26°C treatments for 24h.

Specific points:

Comment 9: In general, the manuscript is well-written. However, additional language is editing to eliminate typos and, especially, to improve grammar.

Response: Thanks for this comment. We are sorry for the editing mistakes. We had corrected and improved the grammar in the revised manuscript.

Comment 10: Considering the importance of *IRT1*, *FRO2*, *AHA2* and the coumarin-related genes *F6'H1*, *S8H* and *CYP82C4* in root Fe uptake, the expression of these genes should be also presented.

Response: As this reviewer suggested, we showed the expression of *FRO2* and *AHA2* in the revised Figure 1A. However, *IRT1*, *F6'H1*, *S8H*, and *CYP82C4* were mainly expressed in roots, thus the expressions of these genes were almost undetectable in our transcriptome data.

Comment 11: Please describe more clearly if whole plants or only roots were sampled for transcriptome reported in Fig. 1A.

Response: Thanks for the comment. We now rephrased it as “the leaves of four-week-old plants were sampled for transcriptome assays” in the figure legend of Fig. 1A.

Comment 12: In plant physiology, the unit of nutrient/chlorophyll etc per unit of biomass is “concentration”, whereas “content” is reserved for unit per organ. Please correct in all figures accordingly. Furthermore, indicate if Fe concentration is on fresh or dry weight basis and whether whole shoots or selected leaves (and if so, which ones) were sampled for elemental analysis.

Response: We agree with this reviewer. In the revision, we corrected the “content” as

“concentration” in all figures accordingly. Now, we rephrase the sentence as “Fe concentration is on the dry weight basis and the leaves of four-week-old plants were sampled for iron analysis” in the figure legend.

Comment 13: Fig. 1E: from the plot, it is hard to understand to what comparisons the delta values refer to.

Response: In the revised Fig.1E, we now describe the delta values referring to the percentage decline of chlorophyll concentration at 8°C compared to 22 °C under Fe-replete (100 μ M Fe) or Fe-excess (300 μ M Fe) condition.

Comment 14: What was the statistical test employed in Fig. 6E?

Response: Sorry, we forget to describe the statistical test in the figure legend. A two factor ANOVA with Tukey’s HSD test was employed in Fig. 6E ($p \leq 0.05$, Data are means \pm SD, n=3 biological repeats).

Reviewer #3:

I strongly think that the authors should provide more experiments if they want to propose that RH24 controls ILR3 translation. These and other concerns are described here:

Comment 1: I have some concerns about the experiments carried out with the *bts* mutants. In most cases, the Col-0 and the *bts-2* mutant were grown at 22°C and then subjected to chilling. However, the *bts* mutant seems to have a developmental phenotype at 22°C (Figure 2A). If there are significant differences in development between Col-0 and the *bts-2* mutant at the time of applying the challenge, how can the authors be sure that the differences are due to chilling sensitivity and not because the *bts-2* plants are already smaller when challenged at 4°C? It is established that the developmental defect associated with *bts-2* is not visible at 26°C. To avoid this problem, they could grow the plants at 26°C and challenge them at 4°C.

Response: We agree with this reviewer and noticed this issue. In fact, we examined the phenotype of *bts-2* at multiple combinations, including 26°C for 2 weeks and then subjected to 26°C, 22°C, 8°C, and 4°C for another 2 weeks (see below Figure 7). *bts-2* showed similar phenotypes at chilling stresses for the ones transferred from 26°C and 22°C, indicating that the initial growth condition is not the major case. Because *bts-2* is stunted but it can complete the life cycle at 22°C, please allow us not include the data from 26°C in the revision. Otherwise, it would make the manuscript to be very complicated.

Fig 7. The phenotype of *bts-2* plants. Plants were grown at 26°C for 2 weeks and then transferred to 26°C, 22°C, 8°C, and 4°C for additional 2 weeks.

Comment 2: Since the authors have RNAseq data regarding the *bts-2* mutant at 22°C and 4°C, it would be helpful to generate a figure similar to the one displayed in Figure 1A to highlight the differences in the expression of Fe-responsive genes in the *bts-2* mutant at both temperatures.

Response: Thanks for the comment. We displayed the expression of Fe-responsive genes in *bts-2* in the revised Figure 1A. Now it is stated as “The results showed that the iron man peptides genes (*IMAs*), and bHLH transcription factors *bHLH38*, *bHLH39*, *bHLH100* and *bHLH101* were significantly downregulated in *bts-2* compared with that in Col-0 at 22°C. However, the iron storage-related genes (*FERS*) in *bts-2* were strongly induced at 4°C, suggesting that chilling stress led to iron accumulation in *bts-2*.”

Comment 3: In Figures 6A-B, the control of the RH24-cYFP is missing. In addition, could authors provide more information about the protein (EV) they use in the negative control? Is it localized in the nucleus?

Response: We are sorry for the missing RH24-cYFP. We added the control of RH24-cYFP in the revised Figure 6A-B. The protein (EV) means empty vector tagged with cYFP or nYFP. Without the target protein interactions, there is no signal for EVs.

Comment 4: In Figures 6D-F, the authors observe that in the presence of RH24, ILR3 accumulates, and that this accumulation is reduced in the *rh24* mutant. Since RH24 seems to bind to ILR3 RNA, they hypothesize that this could be due to an enhancement of translation. However, since RH24 and ILR3 proteins interact, it is also possible that in the presence of RH24 and/or cold, the ILR3 protein could be more stable and be degraded by the proteasome in the absence of RH24. I suggest conducting the experiments in the presence of proteasome inhibitors. Additionally, they should directly probe if ILR3 is regulated at the translational level by RH24. There are different procedures, such as polysome gradients, sucrose cushions, and ribo-Seq, that could be done in the wild type and *rh24* mutant at 22°C and 4°C. A genome-wide translation study on the *rh24* mutants could also provide additional targets of RH24 translational regulation. These experiments could also be done to assess how RH24 accumulates under chilling conditions, since RH24 accumulates at 4°C, but its transcription doesn't seem to change, RH24 could also be a good candidate to be regulated at the translational level.

Response: Thanks for the comments. We now included the protease inhibitor MG132 in the experiments, and we did not see much accumulation of ILR3, suggesting that RH24 and/or cold increases ILR3 protein levels not by reducing ILR3 degradation (see the revised Figure 6D and 6F). To investigate whether it increases the translation, we choose the in vitro translation kit (TNT Quick-coupled transcription translation system, Promega, L2080) that works well in our hands, but we did not see the difference, which also indicates that RH24 regulates ILR3 protein at post-transcription levels. To further explore the mechanism, we analyzed the transcriptome of *rh24* and RH24-ox lines by

the integrated genome (IGV) browser. We discovered that *ILR3* displayed mis-spliced transcripts in *rh24* mutants (see below Figure 8A), which was also seen in RH24 homologies (*AtRCF1/AtRH42*, *OsRH42*, and *PRP5*) that were involved in pre-mRNA splicing (Mingam et al., 2004, *Plant Biotechnology Journal*; Guan et al., 2013, *Plant Cell*; Lu et al., 2020, *Plant Physiology*; Zhang et al., 2021, *Nature*). The RT-PCR result showed that the levels of intron-containing isoform of *ILR3* were higher in *rh24* mutants, but lower in the *RH24-ox* lines at 4°C (see below Figure 8B), indicating that RH24 plays a potential role in pre-mRNA splicing. In addition, we did not see the temperature had effect on *ILR3* transcription, suggesting that low temperatures lead to *ILR3* accumulation at post-transcription level.

Fig 8. RH24 regulates pre-mRNA splicing of *ILR3* under chilling stress.

(A) Visualization of the intron retention of *ILR3* in Col-0, *rh24*, and *RH24-ox* plants by the integrated genome (IGV) browser. The exon-intron structure of *ILR3* is displayed at the bottom. The positions of primers used for RT-PCR analysis are indicated by arrows.

(B) Mis-spliced transcript of *ILR3* in in Col-0, *rh24*, and *RH24-ox* plants under 4°C for 24h. Plants were grown at 22°C for 4 weeks and then transferred to 4°C for 24h. The leaves were sampled for the RT-qPCR analysis. Unspliced mRNA is indicated by single arrows

Comment 5: Have the authors included the 5'UTR and/or the 3'UTRs of *BTS* and *ILR3* in the 35S: *BTS*-T7 and p35S: *ILR3*-T7 constructs in Figures 6D-F? It has to be considered that in many cases the translational regulatory elements are present in the UTRs and that these elements could be essential to observe the translation-dependent accumulation of these proteins. Taking into account that RH24 is an RNA helicase and, therefore, could have a role in translation regulation, without these sequences, for example, they cannot fully discard in Figure 6B that RH24 is involved in regulating *BTS* accumulation. Additionally, it might be possible that the addition of these sequences could provide different results regarding the *BTS* and *ILR3* accumulation.

Response: We agree with this reviewer. We constructed and expressed the 5'UTR-CDS-3'UTRs of *BTS* or *ILR3* in *Arabidopsis* protoplast, and checked whether RH24

regulated the accumulation of *BTS* or *ILR3* (see the revised Figures 6B and 6D). The result showed that RH24 was not yet involved in regulating BTS accumulation, but regulated ILR3 accumulation, and is similar to the *35S:ILR3(CDS)*, which rules out the UTR roles in this case.

Comment 6: In Figures 6D, 6F, S4, the authors provide a quantification of a single experiment. Is it possible to quantify and show the statistics of the whole set of three independent experiments? Additionally, I have not seen what mRNA stands for in these figures; I suppose it is a semiquantitative PCR of the ILR3, BTS genes, etc. Authors should provide the statistics of the whole set of experiments and the analysis of expression by qRT-PCRs, if possible.

Response: In the revision, we analyze the expression of *RH24* and *ILR3* by qRT-PCR (see the revised Figure 4F and 6F). Meanwhile, the statistics of the whole set of three independent experiments were quantified and showed in the revised Figure 4F, 6D, and 6F.

Comment 7: Figure 7. To probe that RH24 controls the formation of ILR3 condensates, I suggest analyzing the nuclear condensates containing ILR3 in the *rh24* mutant and RH24 OE plants? In addition, I miss the information about the role of RH24 in Fe sensitivity in the model. Please include it.

Response: We agree with this suggestion. See the response to Reviewer #1's comment 1. It showed that ILR3 was able to locate in the nuclear bodies in the RH24-ox line under 8°C, but not in *rh24* mutant (see above Figure 1C), indicating that RH24 is required for the formation of ILR3 condensates. We now include the role of RH24 in Fe sensitivity in the model.

Comment 8: Have author tested the expression of the ILR3 responsive genes in *rh24* and RH24 OE lines? Since ILR3 is a TF it would be interesting to analyze the expression of the ILR3 responsive genes to further support the hypothesis.

Response: As this reviewer suggested, we performed the expression of the ILR3 responsive genes (*bHLH38*, *bHLH39*, *bHLH100*, and *bHLH101*) in *rh24* and *RH24-ox* lines by the RT-qPCR assays (see below Figure 9). The results showed that the expression levels of ILR3 responsive genes were remarkably decreased in *RH24-ox* lines, indicating that these iron deficiency-induced transcription factors were suppressed in *RH24-ox* lines. We reason that because RH24-OE leads to ILR3 arrested in the condensates (see above Figure 1C), therefore, ILR3 cannot activate the downstream gene responses. Similarly, chilling also facilitates ILR3 condensate formation (Figure 7C and S7C), which leads to the low expression of ILR3-responsive gene in Col-0. All these data are consistent with our overall conclusion.

Fig 9. The expression levels of *ILR3* responsive genes in *rh24* and *RH24-ox* lines. Col-0, *rh24* and *RH24-ox* plants were grown at 22 °C for 4 weeks and then subjected to 4°C for 24 h. The leaves were sampled for RT-qPCR assays.

Comment 9: In Figure S1 E-F, considering the legend, it seems that E and F are different ways to present the level of expression of the CBF-responsive genes in the *bts-2* mutant at control and at 4°C. However, it seems to me that in F, most of the significant changes occur at 0 h, while in E, the most significant changes occur at 3 h and 24 h.

Response: Sorry for the confusion of the figures. In Figure S1E, it showed that there were no significant changes of the selected CBF-responsive genes between Col-0 and *bts-2* at the same time point (3h or 6h), but not for the changes between different time points. We now make it clearly in the legends.

Minor points:

Comment 10: The authors do not follow a logical order to present the data in the Figures. For example, in Figure S1, G panel is between C and E panels, and this also happens in other figures. I think it would facilitate the reading if the panels in the figures are logically presented.

Response: We are sorry for the orders. We improved the order in all the figures.

Comment 11: In Figure 2B, the legend does not clearly indicate whether these specific experiments were carried out under chilling or control conditions.

Response: We are sorry for the legends. The experiments were carried out under chilling conditions. It's been corrected now.

Comment 12: Some paragraphs are difficult to follow. For example, the paragraph in lines 159-165. (Fig 1A)

Response: We now rephrased the sentences.

Comment 13: When referring to 4°C conditions, I would rather use chilling sensitivity than temperature sensitivity.

Response: We agree with this reviewer. We now rephrased it as “chilling sensitivity” in all sentence and figures accordingly.

Comment 14: In line 250, there should be a mistake with the panels in the Figure S3, since figure S3A does not show the chilling and iron deficiency phenotypes of *bts-2*.

Response: Thanks for this comment. It’s been corrected.

Comment 15: For clarity, in the description of the *bts-r* lines expressing RH24 (DAAD), I would rather refer to “phenocopy the *bts-2* mutant.”

Response: We agree with this reviewer’s comment. We now re-state as “phenocopy the *bts-2* mutant”.

Reviewer #4:

Comment 1: Figure 1A and Figure S1E. The authors show the expression of the iron metabolic pathway DEGS. In line with the results, they observed high expression of BTS in Col-0 plants at 26 and 4 °C, and a different pattern of CBF genes between Col-0 and *bts-2* mutant. In base on these results, one would expect to observe a differential expression for *ILR3* and the helicase gene family under the same conditions. Have the authors looked at these specific genes?

Response: We analyzed the pattern of *ILR3* in Figure 1A and some helicase genes in below Figure 10. Probably because the transcription levels of *ILR3* were not significantly influenced by chilling stress (see above Figure 5B), the *ILR3* mRNA abundance has no remarkable difference between Col-0 and *bts-2* at 4 °C. Some homologous genes of *RH24* were slightly induced at 4°C in Col-0 and *bts-2*. However, we prefer not including these data in the revision as it would confuse readers. In fact, *RH24* was discovered in our case by suppressor screening and the transcriptional regulation is not important for either *RH24* or *ILR3* under chilling condition.

Fig 10. The expression of helicase genes in Col-0 and *bts-2* mutants under chilling stress.

Col-0 and *bts-2* plants were grown at 22 °C for 4 weeks and then subjected to 4°C for 24 h. The leaves were sampled for transcriptome assays.

Comment 2: Figure 1D. Based on this result, it is rather clear that *bts-2* displays an increased sensitivity to cold (above 0 °C). Due to the stunted growth of this mutant at 8 °C, it is hard to see an effect of the iron concentration on the mutant growth. I think showing the same experiment with an additional cold acclimation period would help to understand these results.

Response: We agree with this reviewer's comment. We now show the experiment with an additional cold acclimation period (2 weeks) in revised Figure 1D. It clearly showed that high iron supply impaired the chilling tolerance in plants, especially for *bts-2*.

Comment 3: Line 189-190: Throughout the manuscript, the authors used the term "enhanced temperature sensitivity." I recommend replacing "enhanced" with "increased."

Response: We agree with this comment. The "enhanced temperature sensitivity" has been changed to "increased temperature sensitivity" as suggested.

Comment 4: Figure 4F and Figure 6F: In general, the quality of these images is low. I strongly recommend showing the protein concentration levels of at least three independent replicates showing the error bars. Additionally, to support the authors' conclusion that RH24 accumulates under chilling conditions in Figure 4F, I suggest including an additional western blot using cycloheximide to support the authors' conclusion that RH24 accumulates under chilling conditions.

Response: Thanks for the comment. We added the cycloheximide treatment in the experiments and showed the protein concentrations in revised Figure 4F and 6F with three independent replicates. RH24 protein levels were significantly accumulated at 4°C, supporting our conclusion that RH24 is accumulated under chilling conditions.

Comment 5: Figure 7F. I found the scheme hard to understand and interpret. Moreover, I was a bit surprised that the final model does not integrate together cold and iron homeostasis

Response: In the revision, we improved the working model with the iron homeostasis in it.

Dear Editor and Reviewers:

Thanks for the valuable feedback. We have carefully addressed all the comments raised by the reviewers. The necessary corrections have been made in red in the revised version of the manuscript.

Reviewer comments:

Reviewer #2:

The authors have addressed all points raised in my initial review, providing new experimental data and clarifications in the text. The new data and the amendments in the text have improved the manuscript. However, some points were not yet addressed satisfactorily and need further clarifications and even more experimental evidence, as outlined below.

Major points:

Comment 1: To reply to my comment of whether the increased cold sensitivity of the *bts-2* mutant simply due to the higher Fe ROS accumulation in leaves, the authors performed one experiment in which they show that supply of ascorbic acid did not rescue the cold sensitivity of *bts-2*. Furthermore, they refer to other studies to support their conclusion that the chilling sensitivity of *bts-2* was not due to the high ROS accumulation in this mutant. This is not completely satisfactory: 1) ascorbic acid can also act as a pro-oxidant, as it can reduce Fe(III), thus evidence based solely on this compound is difficult to interpret; and 2) the conditions used in the mentioned studies with H₂O₂ supply, are not necessarily reflecting the ROS accumulation in plant tissues induced by the Fe overload in *bts-2*. Actually, there is an alternative way to directly address this critical question, which the authors did not explore. Most phenotypes of *bts-2*, including the strong Fe overaccumulation, are almost fully reverted by growing plants at 26°C (Figs. 1C and S1). Is ROS accumulation decreased at this elevated temperature? If yes, would *bts-2* plants pre-cultured at 26°C still show increased chilling sensitivity when then transferred to lower temperatures?

Response: Thanks for this insightful comment. We recognize that ascorbic acid has pro-oxidant effects, which is usually at higher concentrations (Kapsokefalou and Miller, 2007, British Journal of Nutrition). However, low concentration of ascorbic acid often acts as ROS-scavengers under various abiotic stresses in plants. In our case, the application of physiological concentration of ascorbic acid (Kka et al., 2017, Plant Growth Regulation) could not revert the chilling sensitivity of *bts-2*, suggesting that accumulation of ROS is not the major reason to cause the sensitivity at least. Another evidence is that the ILR3-ox plants accumulate more iron but do not accumulate much ROS, and they were sensitive to chilling stress. However, it is difficult to decouple ROS damage completely from iron accumulation under chilling stress, as iron is required for Fenton reaction. Therefore, it is safe to conclude that in our case iron accumulation is a major reason to make the plants vulnerable to low temperatures.

As suggested, we checked the ROS accumulation in *bts-2* at 26°C. By DAB staining, we observed that the ROS was not accumulated in *bts-2* (see the below figure). In the revised Figure S1E, we demonstrated that *bts-2* plants pre-cultured at 26°C still exhibit increased chilling sensitivity when transferred to lower temperatures. These results also

suggested that the chilling sensitivity of *bts-2* is, to a great extent, not mainly due to the ROS accumulation.

The DAB staining of *bts-2* plants. Col-0 and *bts-2* plants were grown at 26°C for 4 weeks, and then the leaves were stained with DAB.

Comment 2: Following on that, can 26°C largely rescue the phenotypes of *bts-2* because it strongly suppresses RH24 protein accumulation at this temperature? This possibility should be at least discussed in the manuscript.

Response: Thanks for this suggestion. It is true that 26°C (higher temperature) strongly inhibits RH24 protein accumulation, which led to the less expression of ILR3; as a result, it rescues *bts-2* due to less iron absorption. We now discuss it in line 308-310.

Comment 3: The experiment suggested above would also provide additional evidence to support that the increased sensitivity of *bts-2* plants to low temperatures is associated with the Fe overaccumulation phenotype and not with another process regulated by the putative E3 ligase function of BTS.

Response: Yes, it is. As in our case, ILR3 is largely regulated by RH24 at post transcription level.

Comment 4: The fact that RH24, ILR3 and BTS are highly expressed in young leaves is very important for the whole model. However, I could not find this dataset in the manuscript. These results must be described and discussed in the manuscript.

Response: We agree. Considering the importance of this result for the whole model, we now show it in the revised Figure S6F, and we have described and discussed it in line 371-373 and line 522-523.

Minor points:

Comment 5: Following on points 1 and 2, does the reversion of several phenotypes of *bts-2* at 26°C suggest that the mutant is not more sensitive to chilling but actually even to normal ambient temperature? This point should be discussed.

Response: Yes, *bts-2* mutant is not merely sensitive to chilling but actually even to normal ambient temperature. We now discuss it in line 530-532.

Comment 6: Some statements do not reflect what is shown by the data. For instance, lines 177-179, from the heatmap of Fig. 1A, only FER1 was “strongly induced” in *bts-2* compared to WT at 4°C and not FERs in general, as the sentence suggests. Please amend

throughout the manuscript.

Response: Thanks for the careful checks. We are sorry for the imprecise statements. We now re-state it as "*FER1*" in line 182.

Reviewer #3:

Comment 1: I consider that authors have answered successfully to my main concerns. However, I feel that they can include in the supplementary data the figure 7 from the answer to the reviewers. I do not see how this could make more complicated the article. Regarding the role of RH24, I feel that it is not completely revealed. However, I also consider that authors have obtained many and important pieces of information about the role of BTS, RH24 and ILR3 in cold stress and iron homeostasis and of the link between the three proteins and the two process. In my opinion, all this information makes this manuscript worth to be published in this journal.

Response: We sincerely thank this reviewer's insightful comments. As this reviewer suggested, we have included this mentioned result in the revised Figure S1E.

Comment 2: Line 491, it is established "Instead, RH24 binds with and likely splices ILR3 mRNA, which consequently regulates ILR3 protein levels". If I am not wrong, the data regarding the role of RH24 in splicing are only included in the section: response to the reviewers, but this information is not available in the Figures to the readers. Without giving this information to the readers, I think that the sentence in line 491 could not be stated. I propose the authors to include the figure and discuss how the splicing might modulate the accumulation of the protein. I also propose the authors to include a sentence to clarify that the role of RH24 in ILR3 regulation of translation cannot be completely excluded.

Response: Thanks for this advice. As suggested, we have included the data in the revised Figures S6I and S6J and described it in line 397-405. Previous studies (Gudikote et al., 2005, Nature Structural and Molecular Biology; Ma et al., 2008, Cell) have demonstrated that the pre-mRNA splicing may enhance the translation efficiency of spliced mRNAs and promote protein production. We discussed it in line 509-511. We also discussed that the role of RH24 in ILR3 regulation of translation cannot be completely excluded in line 514-516.

Minor suggestions:

Comment 3: Line 79, "Except for the transcription factors, post-transcriptional regulation is also involved in chilling tolerance", I suppose that authors want to say "As long as transcription, post-transcriptional regulation is also involved in chilling tolerance.."

Response: Thanks for this advice. We now rephrase it as "In addition to the transcription factors, post-transcriptional regulation is also involved in chilling tolerance" in line 80.

Comment 4: Line 163 "DEGS of Col-0 plants showed remarkably reduced expression for the major iron-responsive genes, including IMAs.....". This sentence will be clearer to me if instead of major iron-responsive genes they include the categories that appear in the figure such as "low iron inducible genes, iron deficiency-induced transcription factors, etc".

Response: Thanks for this comment. We now rephrase it as "DEGS of Col-0 plants

showed remarkably reduced expression for the genes involved in regulating low-iron-inducible iron homeostasis, encoding iron deficiency-induced transcription factors and low-iron-inducible ferric chelate reductase at 4°C” in line 168-170.

Comment 5: What do mean rh24#5 sgRNA and rh24#2 sgRNA in Figure 3C?

Response: Sorry, we forgot to describe it in the figure legend. In Figure 3C, the rh24#5 sgRNA and rh24#2 sgRNA refer to the position of two sgRNA:Cas9 targets of RH24 (see Figure S5F). Now, we rephrase it as “*sgRNA-RH24-5*” and “*sgRNA-RH24-2*” in the revised Figure 3C. We also describe it as “The positions of two sgRNA:Cas9 targets of RH24, T-DNA insertions, and the nucleotide substitutions in *bts-r* are shown” in the figure legend of Figure 3C.

Comment 6: Figure 4. Authors make a lot of effort to demonstrate that RH24 is an ATP dependent RNA helicase. Could they include in the discussion how this activity could modulate the splicing of ILR3?

Response: A number of studies have been reported that the DEAD-box RNA helicase RH24 homologies may function in pre-spliceosome formation through an ATP-dependent function to remodel U2 small nuclear ribonucleoprotein particles (snRNPs), and proofreading of the branch site sequence, then processing intron removal and exon ligation (Liang and Cheng., 2015, *Genes and Development*; Zhang et al., 2021, *Nature*). We now discuss it in line 507-514.

Comment 7: Line 296 ...” we conclude that the dsRNA unwinding activity of RH24 is indispensable for suppressing chilling sensitivity to *bts-2*”. I would soften the sentence to “we conclude that, most probably, the dsRNA unwinding activity of RH24 is indispensable for suppressing chilling sensitivity to *bts-2*”.

Response: We thank this suggestion, and now we soften this sentence and rephrase it in line 300.

Comment 8: Line 340. I think that there is a mistake and they do not refer in this paragraph to Figure 5E.

Response: Thanks for the careful checks. We are sorry for this mistake. It's been corrected in line 341-342.

Comment 9: I think that authors should rephrase those sentences related to the effect of MG132: Line 377, line 390 and 493, they are quite confusing. For example, in the case of phrase, line 390, “the proteasome inhibitor MG132 did not increase the accumulation of ILR3 in the ILR3-ox/rh24-1 plants, suggesting that RH24 facilitates ILR3 accumulation at transcription or translation level under chilling conditions”, I would propose “the proteasome inhibitor MG132 did not increase the accumulation of ILR3 in the ILR3-ox/rh24-1 plants, suggesting that the reduced accumulation of ILR3 accumulation in the rh24-1 is not due to ILR3 protein degradation”.

Response: Thanks for this comment. We are sorry for these confusing sentences. In the revised line 378, we now rephrase the sentence as” suggesting that ILR3 protein is

stable in the presence of RH24". In the revised line 392, we now rephrase the sentence as "... suggesting that ILR3 protein levels were dependent on RH24 and RH24 seems to stabilize ILR3". In the revised line 504-506, we now rephrase the sentence as" which consequently regulates ILR3 protein levels although it also seems to stabilize ILR3".

Comment 10: Line 491 ", and bts-2 chilling sensitivity is suppressed by RH24 (Figures 3 and S6)". Do you mean: ", and bts-2 chilling sensitivity is suppressed in the absence of RH24 (Figures 3 and S6)"?

Response: Thanks for this comment. Now, We rephrase the sentence as ", and bts-2 chilling sensitivity is suppressed in the absence of RH24" in line 502.

Comment 11: I observed some confusing sentences in the materials and methods. For example line 690..." These constructs were mixed and transformed into *N. benthamiana* leaves by *Agrobacterium tumefaciens*,". I suppose that they want to say "These constructs were used to transform *Agrobacterium tumefaciens*,..., and the transformed strains were mixed to agroinfiltrate *N. benthamiana* leaves. This could be also read in line 693. If so, please, could go through materials and methods to describe properly the experiments.

Response: Thanks for this comment. We are sorry for the confusing sentences in the materials and methods. We have carefully reviewed the materials and methods and described properly. Now, in the revised line 711, we rephrase the sentence as "the indicated nLUC and cLUC constructs were transformed into *Agrobacterium tumefaciens* strain C58C1, and then the strains were mixed at a final OD600 of 0.4 and were infiltrated into *N. benthamiana* leaves". In revised line 718, We also rephrase the sentence as "These constructs were transformed into *A. tumefaciens* strain C58C1, and transiently expressed in *N. benthamiana* leaves for 48 h."

Reviewer #4 (Remarks to the Author):

Reviewer #2:

Comment 1. L159-160: Please, rephrase to: Because BTS is a key regulator of Fe deficiency responses, ...

Response: We have rephrased it in Line 159-160.

Comment 2. L416: Correct abbreviation to FRAP

Response: It's been corrected.

Comment 3. In several plots, the dots representing the original datapoints are too small. Actually, the size of the dots is not consistent, sometimes not even within the same plot (e.g., Fig. 2F). I recommend the authors to choose a dot size that allows to clearly see the datapoints (e.g., plot of Fig. 6F) and use that consistently in all plots.

Response: We now we use a suitable dot size that consistently in all plots.

Reviewer #3:

Minor comments:

Comment 1: I strongly recommend that the authors include the number of plants used per replicate in their experimental design.

Response: We have included the number of plants used per replicate in the legends of Figure 1C,1E,2F,2G,5D,5E,7E.

Comment 2: In Figure 2 C, it is stated that "...and the iron deficient condition was made by applying 300 μ M Ferrozine in $\frac{1}{2}$ MS medium to chelate iron (right panel) ..." I believe that the authors intended to refer to the lower panel instead of the right panel.

Response: It's been corrected.